# Dynamic properties of noise and Her6 levels are optimized by miR-9, allowing the decoding of the Her6 oscillator

Ximena Soto[1,*,†] (ID), Veronica Biga[1,†], Jochen Kursawe[2], Robert Lea[1], Parnian Doostdar[1], Riba Thomas[1] & Nancy Papalopulu[1,**] (ID)

## Abstract

Noise is prevalent in biology and has been widely quantified using snapshot measurements. This static view obscures our understanding of dynamic noise properties and how these affect gene expression and cell state transitions. Using a CRISPR/Cas9 Zebrafish *her6::Venus* reporter combined with mathematical and *in vivo* experimentation, we explore how noise affects the protein dynamics of Her6, a basic helix-loop-helix transcriptional repressor. During neurogenesis, Her6 expression transitions from fluctuating to oscillatory at single-cell level. We identify that absence of miR-9 input generates high-frequency noise in Her6 traces, inhibits the transition to oscillatory protein expression and prevents the downregulation of Her6. Together, these impair the upregulation of downstream targets and cells accumulate in a normally transitory state where progenitor and early differentiation markers are co-expressed. Computational modelling and double smFISH of *her6* and the early neurogenesis marker, *elavl3*, suggest that the change in Her6 dynamics precedes the downregulation in Her6 levels. This sheds light onto the order of events at the moment of cell state transition and how this is influenced by the dynamic properties of noise. Our results suggest that Her/Hes oscillations, facilitated by dynamic noise optimization by miR-9, endow progenitor cells with the ability to make a cell state transition.

**Keywords** cell state transitions; gene expression noise; Her6 oscillations; miR-9; Zebrafish neurogenesis

**Subject Categories** Development; Signal Transduction

**The EMBO Journal (2020) 39: e103558**

## Introduction

Understanding cell state transitions is important for the mechanistic understanding of development, regeneration and cancer. Gaining this understanding with single cell level resolution in real time and in the tissue environment is the ultimate challenge. Here, we aim to provide this understanding for cell state transitions that take place in the development of the nervous system.

Transcriptomic analysis of single cells is currently the standard used to understand cell state transitions as it can detect thousands of genes that are up- or downregulated in each state. Aided by machine learning, it can infer the most likely path taken by cells, the branching points of differentiation, and reconstruct the temporal order of cell states ("pseudotime" as in Farrell *et al*, 2018; Sagner *et al*, 2018). However, it is a snapshot method that cannot reveal the fine-grained dynamics of gene expression in a timeline of just a few hours. Other snapshot methods such as reporter expression variability across a population have been used to infer single cell dynamics but make certain assumptions, such as ergodicity of the system (e.g. Kalmar *et al*, 2009) and homogeneity of dynamic cellular behaviours across the population, both of which may deviate from reality, as recently shown in Manning *et al* (2019). Even with improved methods of incorporating prior factors of interest in such methods (Campbell & Yau, 2018), biological noise analysis is by necessity restricted to quantifying the variability in the data (Eling *et al*, 2019) and cannot provide information on the dynamic properties of noise.

Are short timescale dynamics, such as gene expression noise and oscillations, important for cell state transitions? Gene expression noise, defined as stochastic events in transcription and translation as a composite of intrinsic and extrinsic noise, is prevalent and whether its function is beneficial or detrimental is both fascinating and hotly debated (reviewed in Balazsi *et al*, 2011; McDonnell & Ward, 2011; Eling *et al*, 2019). In parallel, evidence is mounting that pulsatile or oscillatory expression that takes place in an ultradian scale carries information encoded in its characteristics, which can

1  Faculty of Biology Medicine and Health, School of Medical Sciences, The University of Manchester, Manchester, UK
2  School of Mathematics and Statistics, University of St Andrews, St Andrews, UK
   *Corresponding author. Tel: +44 0161 2757221; E-mail: ximena.soto@manchester.ac.uk
   **Corresponding author. Tel: +44 0161 306 8907; E-mail: nancy.papalopulu@manchester.ac.uk
   †These authors contributed equally to this work as first authors

be decoded by downstream processes (reviewed in Levine *et al*, 2013). Although the benefits of pulsatile gene expression are recognized in many systems across biology, so far, examples in developmental biology are few. Leading developmental examples are the ultradian oscillations of transcription factors (TFs) and signalling molecules in the context of somitogenesis and neurogenesis. Key examples include *hes* genes and proneural TFs, e.g. Ascl, Olig and Ngn, members of Notch signalling (e.g. delta, Imayoshi *et al*, 2013; Shimojo *et al*, 2016, 2008; reviewed in Kageyama *et al*, 2019) and Wnt signalling pathways (Sonnen *et al*, 2018). Wherever it has been tested by experimentation, it was clear that sustained versus pulsatile expression of such molecules has distinct outcomes for cell fate decisions (Nandagopal *et al*, 2018) reviewed in Kageyama *et al* (2019).

For the development of the nervous system, understanding the dynamics of *hes* gene expression is particularly important because TFs of this family are described as being important for neural progenitor maintenance and controlled differentiation (Hatakeyama *et al*, 2004). Oscillatory dynamics can be revealed with live imaging using protein reporter fusions whose value is that they are more likely to recapitulate the properties of the endogenous proteins, many of which are highly unstable. Indeed, instability of components (mRNA or protein) is an essential property in several biological oscillators (Novak & Tyson, 2008). In neurogenesis, such protein fusions have been invaluable in characterizing the oscillatory dynamics of *hes* genes, proneural genes (*ascl*, *ngn* and *olig2*) and *dll* (reviewed in Kageyama *et al*, 2019). However, with few exceptions (e.g. Shimojo *et al*, 2016), most of these studies have been performed in dissociated cells, cultured in 2D. Recent evidence in mouse *ex vivo* sections suggests that the tissue environment can modify the oscillatory dynamics (Manning *et al*, 2019); therefore, it is essential to be able to study protein expression dynamics *in vivo*. Furthermore, these mouse studies also suggested that cell state transitions may be noise-driven. Thus, it is essential to understand how biological noise affects the performance and decoding of an oscillator and how such noise may be controlled *in vivo*.

Zebrafish is ideal for such *in vivo* studies because of its superior suitability for live imaging of molecular and cellular events at several timescales. This has been exploited in the context of oscillations during somitogenesis, both at the population and single-cell level (Soroldoni & Oates, 2011; Delaune *et al*, 2012; Webb *et al*, 2016). Previous studies based on fixed tissue snapshot analysis suggested that *her* genes maintain cells in an ambivalent progenitor state, controlled by miR-9 (Leucht *et al*, 2008; Coolen *et al*, 2012), consistent with the findings that miR-9 targets *hes1* in the mouse (Bonev *et al*, 2012; Goodfellow *et al*, 2014; Phillips *et al*, 2016). However, nothing is known about the real-time dynamics of gene expression and the associated noise *in vivo* during Zebrafish neurogenesis.

Here, we use CRISPR/Cas9 technology to create the first fluorescent moiety knock-in Zebrafish to be used beyond proof of principle (Kesavan *et al*, 2017) for experimental purposes. We knocked-in Venus fluorescent protein in frame with Her6, a Hes1 homologue, and after thorough characterization of the reporter, we used it to study the endogenous Her6 protein dynamics in Zebrafish hindbrain neurogenesis. We find that Her6 is initially expressed in neural progenitor cells (NPCs) in a fluctuating and noisy but aperiodic manner. Oscillations with regular periodicity are observed at the peak of neurogenesis and coincide with the onset of expression of miR-9 in the hindbrain, consistent with previous articles reporting post-transcriptional targeting of *hes1/her6* by miR-9 (Bonev *et al*, 2012; Coolen *et al*, 2012). To investigate the precise function of miR-9 in Her6 dynamics, we use CRISPR/Cas9 to mutate the miR-9 binding site in the 3′UTR of *her6*. We report that preventing the influence of miR-9 on Her6 increases the amount of protein expression noise, prevents the downregulation of Her6 protein levels and decreases the number of oscillators during neurogenesis. The noise increase is characterized by high frequency and by analogy to a concept in Engineering and Neuroscience where noise inhibits rhythmic phenomena (Uzuntarla *et al*, 2013; Bacic *et al*, 2018), and we term this (molecular) inverse stochastic resonance. Our theoretical framework and smFISH analysis suggest that increased frequency noise interferes with the decoding of the Her6 oscillator by preventing the upregulation of downstream genes and the downregulation of Her6 expression levels. Finally, we confirm experimentally that under these conditions, cells fail to downregulate Her6, to upregulate proneural genes and to differentiate; instead, they accumulate in a transitory state, which co-expresses progenitor and early neuronal markers. Together, these results suggest that the function of Her6 oscillations is to allow progenitors to make a cell state transition to differentiation and this can be either facilitated or impeded by noise, depending on its dynamic properties.

# Results

### A *her6::Venus* knock-in protein fusion is a quantitative and faithful reporter of endogenous Her6 protein dynamics

In order to characterize the dynamics of cell state transitions, we aimed to identify the most suitable Zebrafish *her* gene for dynamic analysis of gene expression. There are two *hes1*-related genes in Zebrafish: *her6* and *her9* (Zhou *et al*, 2012). They are both expressed in the Zebrafish embryonic central nervous system (CNS; Fig 1A, Appendix Fig S1A and B) mostly in a mutually exclusive pattern appearing as adjacent narrow "bands" of cells that span the dorso-ventral axis in the hindbrain (Fig 1A). Both *her6* and *her9* harbour a miR-9 binding site in the 3′UTR, but the *her6* site is a better quality-binding site (7A1-mer rather than 6-mer; Appendix Fig S1C); therefore, we decided to focus on *her6*.

To generate a reporter that would be suitable for live imaging, we devised a CRISPR/Cas9 knock-in strategy that would preserve the properties of the endogenous protein (Fig 1B, Appendix Fig S1D-I). We carried out a number of tests to ensure that the reporter recapitulated accurately the expression of Her6 (Appendix Fig S2A-I). First, the expression of Venus was compared to the expression of endogenous Her6 by chromogenic and fluorescent whole-mount *in situ* hybridization (WM-ISH) and sections through the hindbrain. Neither ectopic nor any region of missing expression were identified (Appendix Fig S2A and B). There was no significant change in the somite number between control, heterozygous or homozygous *her6::Venus* embryos at 72 hpf (Appendix Fig S2E), suggesting that the knock-in reporter does not interfere with normal development. The protein molecule number was estimated in single NPCs by fluorescence correlation spectroscopy (FCS) in homozygous and heterozygous embryos and the ratio was found to be 1.8, indicating

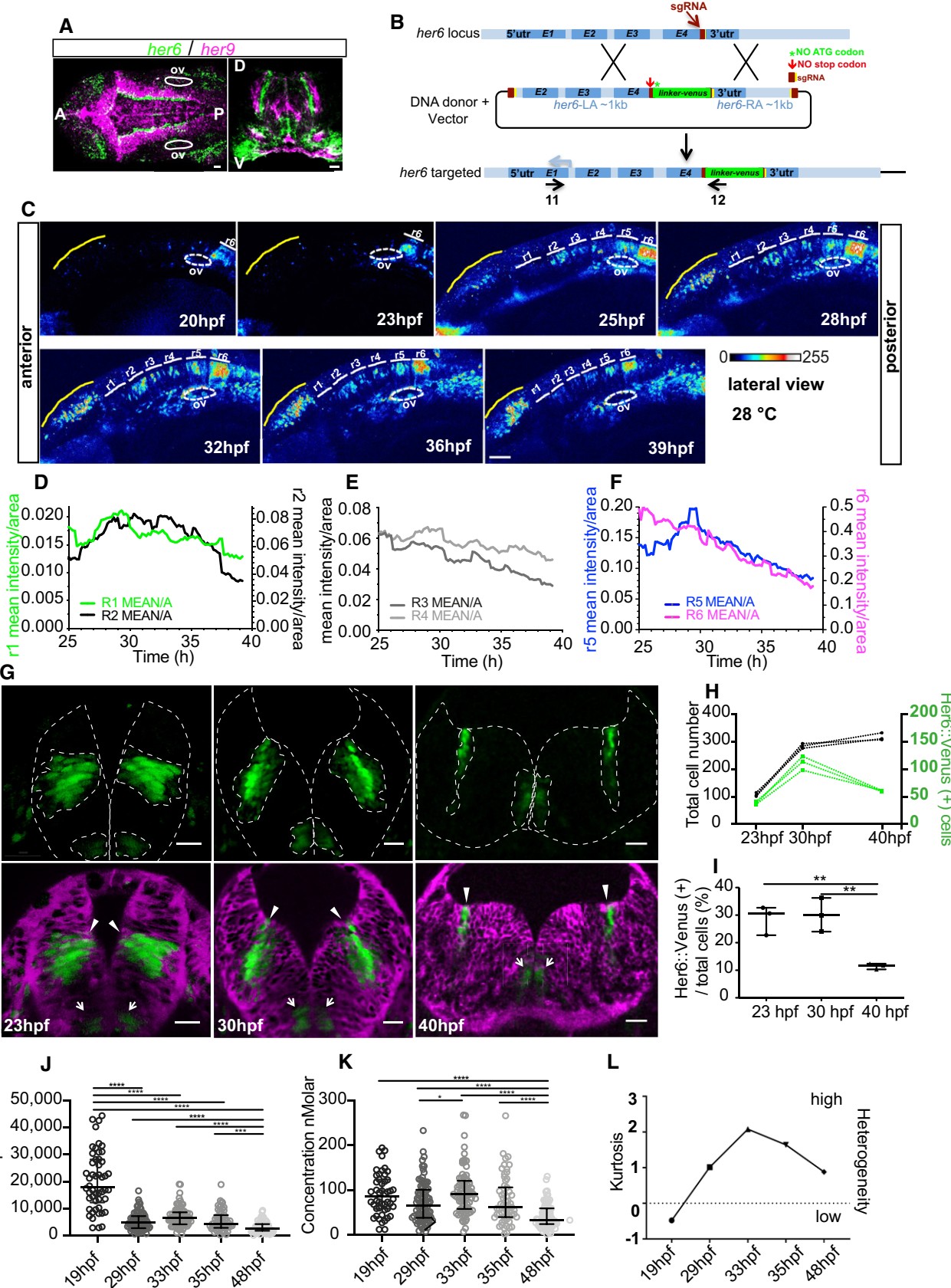

**Figure 1.**

◀

**Figure 1. Her6::Venus protein expression during Zebrafish neural development.**

A    Double-fluorescent whole-mount *in situ* hybridization (WM-ISH) to detect *her6* (green) and *her9* (magenta); coronal view (left panel) and transversal section (right panel), scale bar 20 μm; 30–32 hpf; annotations denote anterior (A), posterior (P), otic vesicle (ov), dorsal (D) and ventral (V).

B    Schematic of strategy used to generate the *her6::Venus* knock-in; left arm, LA; right arm, RA.

C    Representative time series example of Her6::Venus expression during development, in the midbrain and hindbrain. Confocal images represented as 2D maximum projection; longitudinal view; scale bar 50 μm; otic vesicle (ov); also included in Movie EV1. r1: rhombomere 1, r2: rhombomere 2, r3: rhombomere 3, r4: rhombomere 4, r5: rhombomere 5, r6: rhombomere 6.

D–F    Intensity mean of Her6::Venus per rhombomere area over development grouped by expression level, related to the r1-r6 regions in panel (C) : (D) r1 and r2; (E) r3 and r4; (F) r5 and r6.

G    Transversal view of r6 in *her6::Venus* embryos over time; Her6::Venus protein expression domains: a ventral domain (arrows) and a more dorsal lateral domain (arrowheads); the caax-mRFP was used as membrane marker (magenta); scale bars 20 μm; images at 30–40 hpf are maximum projection of 4 z-stacks from Movie EV2.

H    Quantification of Her6::Venus(+) cell number (green) compared to total cell number (black) over development.

I    Proportional changes in Her6::Venus(+) cell numbers during development; bars indicate median and interquartile range of counts collected from 3 different *z*-stacks per embryo and one embryo per condition; one-way ANOVA with Bonferroni multiple comparison test with significance **$P < 0.01$.

J, K    Nuclear abundance and nuclear concentration of Her6::Venus protein in homozygous embryos at different stages during development measured by fluorescence correlation spectroscopy (FCS); bars indicate median and interquartile range of 19 hpf: 6 embryos, 50 cells; 29 hpf: 6 embryos, 86 cells; 33 hpf: 5 embryos, 76 cells; 35 hpf: 6 embryos, 59 cells; and 48 hpf: 6 embryos, 89 cells and Kruskal–Wallis with Dunn's multiple comparison test, significance *$P < 0.05$, ***$P < 0.001$, ****$P < 0.0001$.

L    Quantification of heterogeneity using kurtosis from concentration data in (K); null kurtosis corresponds to a normal distribution.

that additional integrations into the genome are unlikely (Appendix Fig S2F and G). The mean number of molecules in the homozygous fish was 7,000 protein molecules per nucleus, at stage 30–34 hpf, which indicates that Her6 protein is a low abundance protein (Appendix Fig S2G), similar to the mouse Hes1 in NPCs (Schwanhausser *et al*, 2011; Phillips *et al*, 2016). Finally, there was no significant change in the protein half-life of HA::Her6 and Her6::Venus, both of which were very unstable (average half-life 12 and 11 min, respectively; Appendix Fig S2H). These findings confirm that the Her6::Venus fusion protein is a faithful reporter for visualizing endogenous Her6 dynamic expression.

## Changes in Her6 protein expression dynamics are hallmark of active hindbrain neurogenesis

Little is known about the dynamics of *her6* gene expression during neurogenesis since only fixed samples analysed by *in situ* hybridization at stages earlier than 24 hpf have been described before (Pasini *et al*, 2001). Thus, we first focused on characterizing the Her6 protein expression in the hindbrain during the period of development when neurogenesis takes place, i.e. between 20 and 48 hpf (Lyons *et al*, 2003). Still, images of the lateral view from the *her6::Venus* knock-in brain reveal that Her6 is expressed in the hindbrain rhombomeres (r1-r6) with an overall temporal gradient since expression starts in r6/r7 at 20 hpf and spreads anteriorly to r1, r2, r3, r4 and r5 (Fig 1C, Movie EV1). Individual rhombomeres show variable levels of Her6, lower in r1/r2/r3/r4 compared to r5/r6 (Fig 1D and E versus F). Between 25–31 hpf expression in r1/r2/r5 shows upregulation (Fig 1D and F); meanwhile, r3/r4/r6 have fluctuating but steady declining levels (Fig 1E and F, Movie EV1). Importantly, all rhombomeres exhibit protein downregulation over time and this coincides with the inflection point when neurogenesis starts increasing exponentially in the hindbrain (Lyons *et al*, 2003). The Her6 expression profile is higher in r5/r6 with r6 having a constant slow decline over time from 25 hpf, reflecting overall the expected Her6 downregulation as cells differentiate and suggesting a prolonged neurogenesis. We selected rhombomere 6 for further dynamic analysis because, in spite of the overall steady decline, Her6 is highly expressed in r6 and the otocyst can be used as a clear anatomical landmark.

Transversal views of the Her6::Venus expression in the hindbrain (r6) showed expression in two restricted domains, each starting close to the ventricular zone and extending further out towards the basal surface. A ventral domain potentially contributes to motor neuron circuits (Fig 1G, arrows and Movie EV2) (Zannino *et al*, 2014), while the more dorsally located domain, halfway along the D-V axis, is likely to encompass interneuron progenitors (Zannino *et al*, 2014) (Fig 1G, arrowheads and Movie EV2). This dorsal domain was the subject of subsequent investigation. The number of Her6 expressing cells within this domain initially increases in absolute numbers but not the percentage of expressing cells (Fig 1H and I, 23 hpf versus 30 hpf); meanwhile, in the later phases of neurogenesis we noted both an absolute decrease and a proportional decrease in Her6-expressing cells (Fig 1H and I, 30 versus 40 hpf), perhaps reflecting a switch from symmetric (i.e. proliferative) to asymmetric (i.e. neuron-generating) divisions of Her6-positive NPCs. There is no reduction of the Her6 domain area between 23 and 30 hpf despite the apparent morphogenetic movements, while reduction between 30 and 40 hpf (Appendix Fig S2I, 30 versus 40 hpf) is due to actual reduction of NPC numbers (Fig 1H and I, 30 versus 40 hpf). When we quantified the absolute protein molecule number per nucleus by FCS, we observed the highest abundance at 19 hpf (Fig 1J); this is a consequence of a larger nuclear volume observed at this stage (Fig 1G, 23 versus 30 hpf) and no significant difference in concentration was seen (Fig 1K, 19 versus 29 hpf). We also observed a wide heterogeneity in the Her6 concentration at the single cell level that had its peak around 33 hpf (Fig 1L).

In summary, we observe a dynamic Her6 expression at a population level with a declining trend on expression, reflecting NPC differentiation. This encompasses high gene expression heterogeneity that could be due to dynamics at single cell level over time and is studied next.

## Her6 expression undergoes a transition from stochastically noisy to oscillatory as neurogenesis proceeds

To characterize the dynamic expression of Her6 in normal hindbrain development, we used live imaging over 10–12 h of *her6::Venus* homozygous reporter embryos with 6-min intervals (Materials and

Methods—live imaging for single cell tracking). We injected mKeima-H2B and caax-mRPF mRNAs to serve as nuclear and membrane landmarks, respectively, that facilitate segmentation of individual cells (Fig 2A, Appendix Fig S3A, Materials and Methods–single cell tracking). Her6, like its mammalian counterpart Hes1, may generate oscillatory expression in the ultradian scale (i.e. with periodicity of a few hours) due to molecular auto-repression of transcription, coupled with instability of the *her6* mRNA and Her6 protein and influenced by miR-9 (Fig 2B) (Tan *et al*, 2012; Goodfellow *et al*, 2014). Semi-automated tracking of Her6::Venus expressing cells produced Venus and mKeima intensity traces over time (Fig 2D, Materials and Methods–single cell tracking), and these were corrected for bleaching and showed no correlation with position in the Z-axis denoting negligible influence from movement in Z position (Appendix Fig S3C–F).

Analysis of the single cell time series of Her6::Venus showed fluctuations in intensity of expression (Fig 2C, red arrow and Movie EV3), which persisted when corrected for non-specific variability by dividing by the mKeima-H2B signal (Fig 2D). In combination with subtraction of long-term trend (Fig EV1, detrended data), we investigated the presence of ultradian periodicity in further analysis. We interrogated the ability of progenitors to oscillate in Her6 levels over time using a statistical method previously developed to detect periodicity in Luciferase time series (Phillips *et al*, 2017) and subsequently improved for noisy fluorescent data in mouse tissue (Manning *et al*, 2019). Our method uses sophisticated computational techniques to infer parameters of two Ornstein-Uhlenbeck (OU) covariance models $K_{OUosc}$ and $K_{OU}$, which are characteristic of periodic and aperiodic dynamics, respectively (Fig 2E), and are used to classify cells into oscillatory and non-oscillatory with statistical significance (Materials and Methods–dynamic data analysis). Our covariance models include a lengthscale term that describes the rate of decay in correlation between subsequent peaks over time, referred to as periodic lengthscale $\alpha_{OUosc}$, and aperiodic lengthscale $\alpha_{OU}$. A higher lengthscale indicates that subsequent points in a time trace become uncorrelated faster (e.g. a decay in signal autocorrelation), and is therefore used here as a measure of noise in a dynamic trace. In addition to lengthscale, the periodic model also includes a cos wave term and this is characterized by frequency $\beta$ and linked to period, $P = 2\pi/\beta$ Both models also account for the variance of the data which we analyse separately, hence, here was set to $\sigma = 1$.

We used the stochastic $K_{OUosc}$ covariance model to characterize Her6 oscillations at multiple embryonic stages (Fig EV1A). Our analysis showed that the proportion of oscillators increases during development (40% at 28 hpf, versus approx. 80% at 30–34 hpf)

(Figs 2F and EV2, Appendix Fig S4D) with a median period of approximately 1.3–1.5 h in all stages examined (Fig 2G and Appendix Fig S4E). Power spectrum analysis confirmed the presence of a dominant ultradian peak in population analysis of Her6::Venus and Her6::Venus/H2B with higher coherence at 30–34 hpf when compared to nuclear marker mKeima-H2B analysed in the same cells (Fig EV2A and B). Furthermore, coherence levels at stages 30 and 34 hpf, when oscillators are prevalent, were high compared to coherence at 28 hpf (Fig EV2A), when oscillators are fewer, thus providing further evidence of a transition at single cell level from non-oscillatory to oscillatory Her6 over the course of development (Fig EV2B). Given that a large proportion of early stage progenitors were non-oscillatory (Fig EV1B), we then used the aperiodic covariance model, $K_{OU}$, to further investigate dynamics irrespective of ability to oscillate. Interestingly, fluctuations in Her6 expression in early progenitors (28 hpf) were characterized by higher aperiodic lengthscale compared to later stages (30 and 34 hpf) reflecting a decrease in rate of decay in signal autocorrelation over developmental time (Fig 2H, Appendix Fig S4F). This indicated that early progenitors are noisier in their gene expression dynamics compared to later stage progenitors. Consistent with this, the analysis of local coefficient of variation (LCOV denoting local standard deviation of signal over mean, a measure used to quantify noise from snapshot data; Eling *et al*, 2019; Kaern *et al*, 2005) showed that early progenitors have higher gene expression variability than late progenitors (Fig 2I, 28 versus 30–34 hpf).

Taken together, our findings suggest that the increase in neuronal differentiation observed during development of the hindbrain (Lyons *et al*, 2003) is characterized by an increase in the number of cells that show oscillatory Her6 expression and an overall decline in the amount of protein expression noise, measured as a decreased rate of correlation decay (lengthscale).

### Absence of miR-9 regulation prevents normal downregulation of Her6

The conversion from noisy to oscillatory expression as neurogenesis progresses implies a functional role for oscillatory Her6 gene expression. Therefore, we sought to make changes that will interfere or modify oscillatory expression of Her6 by changing the interaction with miR-9. In the Zebrafish hindbrain, miR-9 expression appears at 30–31 hpf, coincident with increased Her6 heterogeneity (Fig 1L) and increased oscillatory expression (Fig 2F), and continues to increase at least until 48 hpf (Fig 3A). The expression of miR-9 and Her6 spatially overlaps, although the expression of miR-9 is wider

---

**Figure 2. Dynamics of Her6::Venus in single neural progenitor cells.**

A   Experimental approach used to image Her6::Venus dynamic expression at single cell resolution.

B   Schematic representation of genetic auto-repression network of Her6 including miR-9 regulation.

C   3D confocal representative images of a single neural progenitor cell (red arrow) tracked over time, data starting from 34 hpf; scale bar 10 μm, images from Movie EV3.

D   Single cell time series of Her6::Venus, mKeima-H2B and Her6::Venus signal normalized by mKeima-H2B corresponding to the cell in (C).

E   Covariance models and parameters used to characterize oscillatory ($K_{OUosc}$) and non-oscillatory ($K_{OU}$) single cell expression.

F–I Quantification of single cell dynamics at different stages in normal development including: (F) proportion of oscillatory cells at different stages in development, (G) period of oscillators estimated using $K_{OUosc}$, (H) noise measured by aperiodic lengthscale ($\alpha_{OU}$) and (I) local coefficient of variation. Dashed lines indicate median, and dotted lines indicate interquartile ranges of 28 hpf (14 cells, 1 embryo), 30 hpf (14 cells, 1 embryo) and 34 hpf (10 cells, 1 embryo); Mann–Whitney two-tailed test, significance *$P < 0.05$, ****$P < 0.0001$.

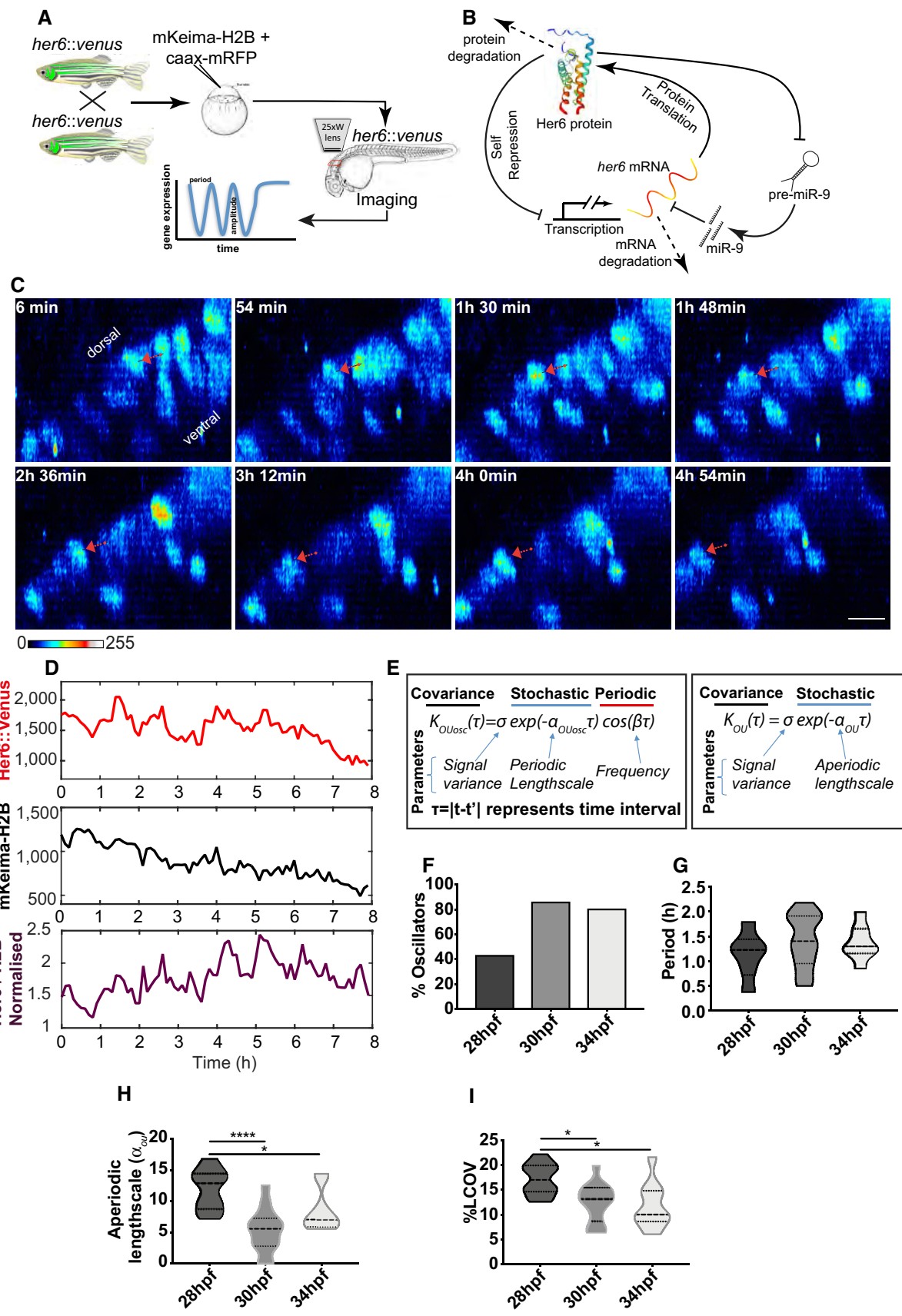

Figure 2.

(Fig 3B), reflecting the existence of other targets (Bonev *et al*, 2011). Based on these findings, we removed the influence of miR-9 on Her6 dynamics by mutating the miR-9 binding site in the *her6:: Venus* 3'UTR, to produce microRNA binding site mutant embryos (MBSm embryos, Fig 3C).

Experimentally, this was achieved by injecting sgRNA targeting the miR-9 binding site together with nuclear cas9 (Cas9nls) protein in one cell stage *her6::Venus* Zebrafish embryos (Appendix Fig S4A). Co-injection of membrane bound mRFP (caax-mRFP) mRNA helped identify the injected embryos and select ones with low mosaicism over the area of interest (Materials and Methods— microinjection and genotyping). High efficiency of mutagenesis of the miR-9 binding site (MBSm) (Appendix Fig S4B and C, with median 80% of injected embryos, each embryo with an average of 8 sequences per embryo, 10 embryos) meant that the phenotypic and dynamic analysis was possible in F0 embryos.

We focused on effects of MBSm starting from 34 hpf when high levels of heterogeneity (Fig 1L) and oscillatory activity (Fig 2F) were observed in normal development. Confocal imaging of live embryos indicated that Her6::Venus expression persisted in MBSm progenitors at stages when expression is downregulated in CTRL (control; inactive sgRNA injected) (Fig 3D and E, Appendix Fig S4G). Using absolute protein quantitation by FCS (Materials and Methods—fluorescence correlation spectroscopy), we showed that the decline in Her6 protein that normally takes place during development does not take place to the same degree in MBSm embryos (Fig 3F and G) and this was also observed in single cell Her6::Venus timelapse data (Appendix Fig S4G). As a result, the median Her6 in MBSm is approximately twofold higher than expected at later stages (Fig 3G).

### Absence of miR-9 regulation on Her6 generates high-frequency protein expression noise

Using our statistical framework (Materials and Methods—dynamic data analysis), we performed a single cell analysis of dynamics in traces observed in paired MBSm versus CTRL *her6::Venus* embryos observed from 34 hpf up to 48 hpf (Figs 4A and B, and EV3). As expected, NPCs from CTRL embryos recapitulated Her6 dynamics observed in embryos without Cas9/sgRNA (Appendix Fig S4D–F) and presence of oscillatory Her6 activity (Fig EV3A). Our analysis of periodicity from detrended data revealed that cells from MBSm

embryos are less frequently oscillatory compared to control (Figs 4C and EV3B), while exhibiting no consistent changes in period (approx. 1–2 h, Fig 4D, Appendix Fig S4D and E). The reduction in oscillatory activity in the absence of miR-9 regulation was consistent with power spectrum analysis (Fig EV4A–E) showing reduced power in MBSm compared to CTRL both at population and single cell level (Fig EV4A and D ultradian). As a consequence, coherence in MBSm embryos was reduced compared to CTRL embryos (Fig EV4C). Coherence in CTRL embryos was high compared to mKeima-H2B (Fig EV4A–C) and analogous to levels observed in uninjected cells (Fig EV2A, 34 hpf). Despite an observed difference in the level of Her6 protein between MBSm and CTRL (Fig 3F and G, Appendix Fig S4G), the quantification of single cell variability showed no differences in local COV (Fig 4E). We then used the aperiodic covariance model, $K_{OU}$, to characterize non-oscillatory activity (Figs 2E and 4F and G) by inferring the rate of correlation decay, i.e. aperiodic lengthscale, $\alpha_{OU}$. We observed an increased rate of correlation decay over time in MBSm compared to control, indicating noisier Her6 expression in the absence of miR-9 regulation (Figs 4H–J and EV3B versus A; $\alpha_{OU}$ values). No change was observed in mKeima-H2B analysed in the same cells (Appendix Fig S4H and I). We also confirmed the effect of increased aperiodic lengthscale using power spectrum analysis (Fig EV4D–G) and identified a significantly increased contribution of high-frequency noise to dynamic Her6 expression in single MBSm cells compared to CTRL (Fig EV4F and G). Thus, the increased aperiodic lengthscale indicates the presence of high-frequency noise in MBSm.

Taken together, our findings indicate that the miR-9 binding site mutation leads to a reduced propensity of neural progenitors to oscillate, failure to downregulate Her6 and increased noise quantified as aperiodic lengthscale.

### Changes in fluctuation dynamics affect downstream expression and cell state progression

To understand how these observed changes in Her6 dynamic expression may affect cell state transitions, we sought to identify how higher levels of noise in the form of increased aperiodic lengthscale may affect the expression of downstream targets. We assumed that being a transcriptional repressor, Her6 (depicted as input Y in Fig 5A) represses the expression of a downstream target X, such as a pro-neural gene, which would mediate the transition to neuronal

---

**Figure 3.  A mutation of the miR-9 binding site affects Her6 level over the course of development.**

A   Chromogenic WM-ISH of miR-9 using miR-9 LNA 5′-Dig observed at different stages during development; longitudinal view, anterior to the left.

B   Transverse section of double-fluorescent WM-ISH for *her6* (green) and *mir-9-4* (magenta) imaged in fixed embryo at 31 hpf; scale bar 30 μm.

C   Schematic representing the miR-9 binding site (MBS) of *her6::Venus* mutated by CRISPR-Cas9nls protein; MBSm refers to specific sgRNA to mutate the MBS as opposed to sgRNA that does not produce mutation (control-CTRL).

D   Confocal imaging of CTRL and MBSm embryo showing Her6::Venus expression in the hindbrain, rhombomeres 3–6 (r3-r6) over the course of development; longitudinal view, scale 30 μm, otic vesicle (ov). Images are representing 2D maximum projection.

E   Her6::Venus expression (red arrows) in hindbrain, rhombomere 6 of live CTRL and MBSm embryos at 52 hpf; edge of ventricular zone shown in yellow; transversal view; scale 30 μm.

F   Her6::Venus protein abundance in CTRL versus MBSm homozygous embryos observed in hindbrain, r6 between 34 hpf and 69 hpf; bars indicate median and interquartile range of 34 hpf CTRL: 36 cells, 1 embryo; 34 hpf MBSm: 35 cells, 1 embryo; 48 ± 2 hpf CTRL: 172 cells, 7 embryos; 48 ± 2 hpf MBSm: 124 cells, 5 embryos; 56 ± 2 hpf CTRL: 69 cells, 4 embryos; 56 ± 2 hpf MBSm: 31 cells, 2 embryos; 69 hpf CTRL: 12 cells, 1 embryo; and 69 hpf MBSm: 6 cells, 2 embryos and Kruskal–Wallis with Dunn's multiple comparison test, significance: *P* < 0.05, ***P* < 0.001, ****P* < 0.0001.

G   Comparative profiles of Her6 protein abundance over development in CTRL and MBSm embryos quantified as median of single cell abundance data in (F).

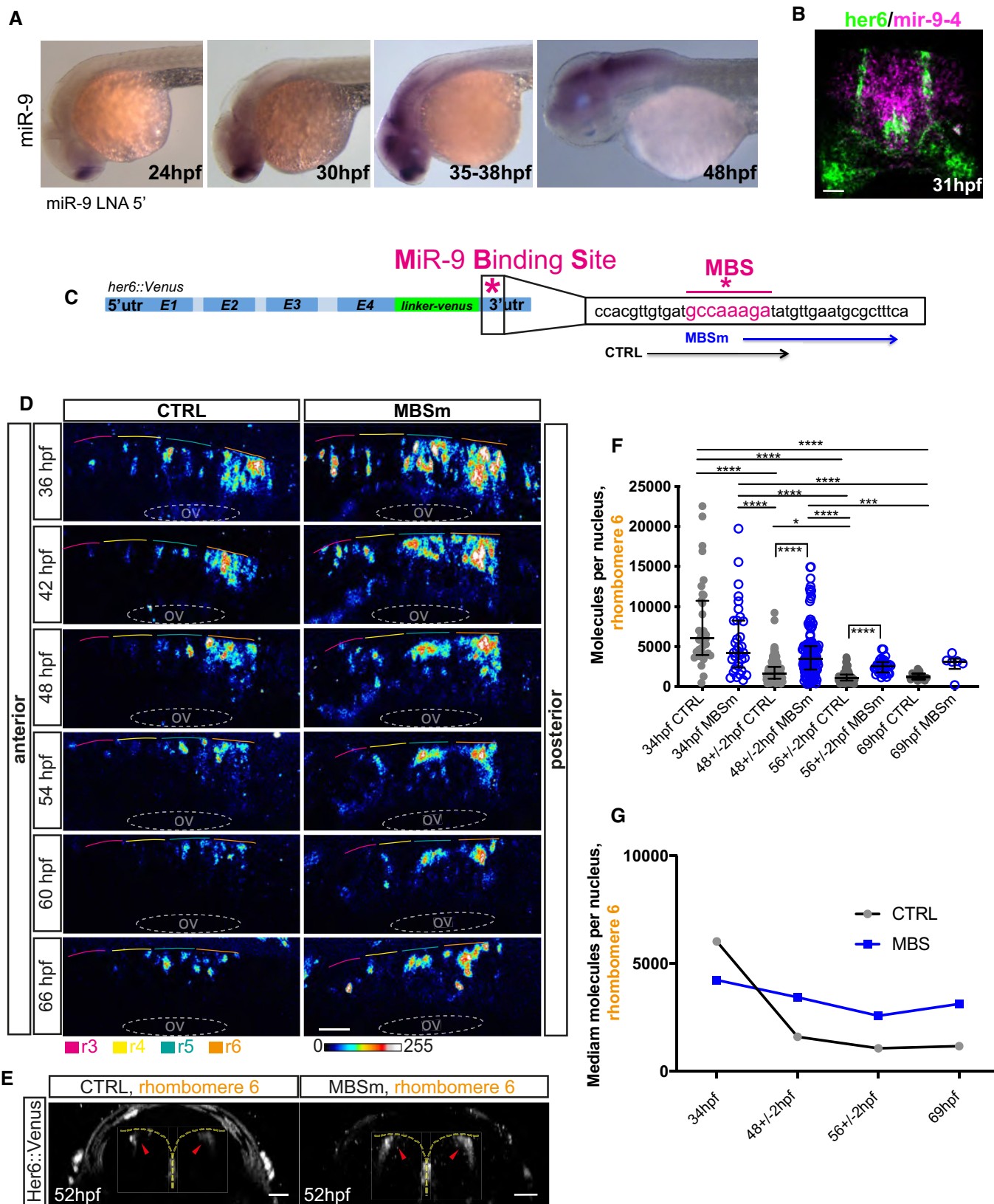

Figure 3.

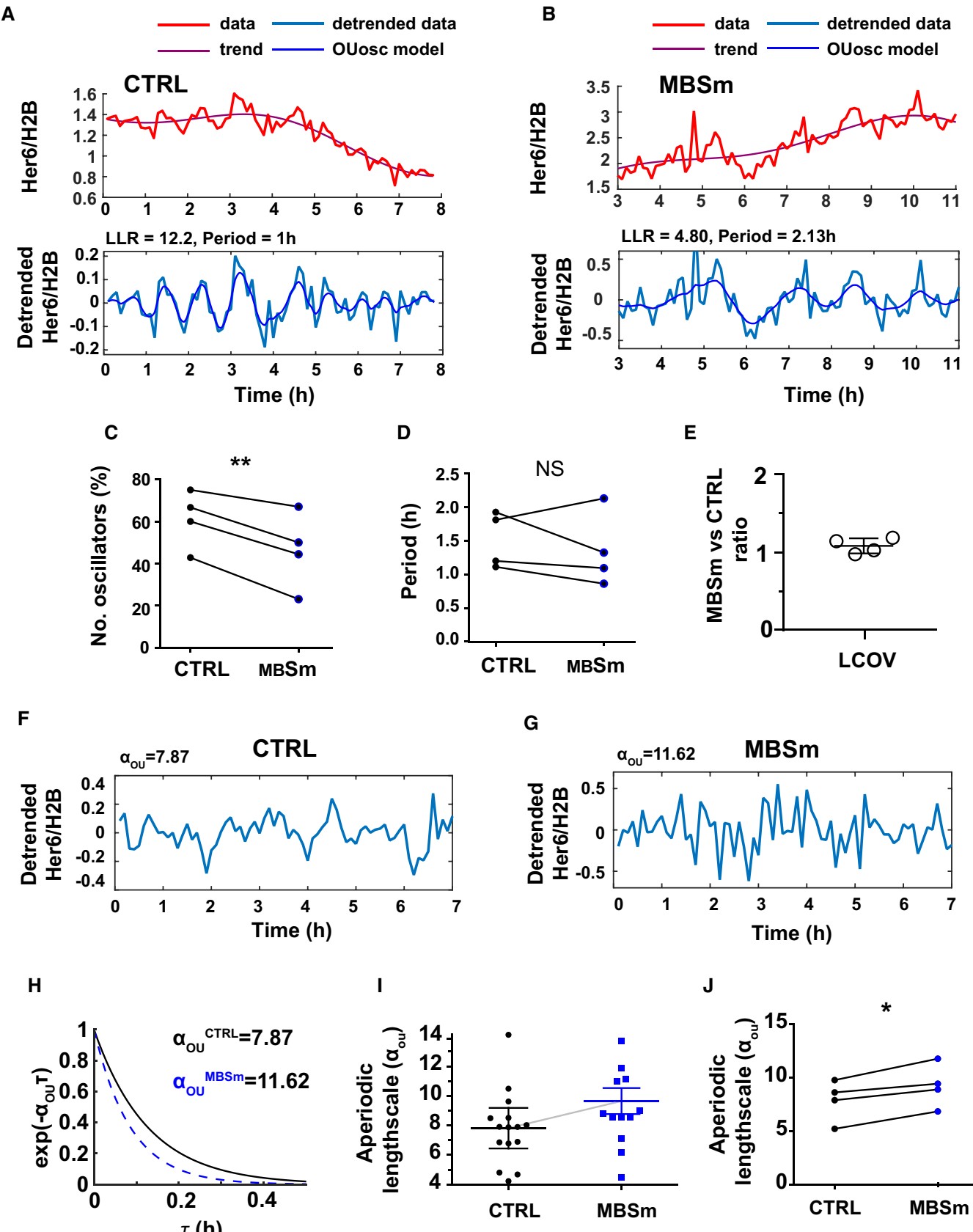

**Figure 4.**

**Figure 4.  A mutation of the miR-9 binding site affects Her6 dynamics at single cell level.**

A, B   Representative examples of single cell oscillators observed in CTRL (A) and MBSm (B) embryos imaged from 34 hpf onwards; time series represent Her6::Venus expression relative to mKeima-H2B (top panel) and detrended relative signal (bottom panel); parameters reported for log-likelihood ratio (LLR) and period correspond to $K_{OUosc}$.

C     Pairwise analysis comparing proportion of oscillators in CTRL versus MBSm embryos indicated as dots; paired *t*-test with two-tailed significance **$P < 0.01$.

D     Pairwise analysis comparing periods observed in CTRL versus MBSm oscillatory cells; dots indicate median per experiment per condition; paired *t*-test with two-tailed non-significance.

E     Analysis of local coefficient of variation (LCOV) as a ratio between MBSm and CTRL; circles indicate median LCOV observed in MBSm cells divided by median LCOV observed in CTRL cells per experiment; bars indicate mean and SEM of LCOV ratios from 4 independent experiments.

F, G   Representative examples of detrended Her6::Venus relative to mKeima-H2B in non-oscillatory single cells observed in CTRL (F) and MBSm (G) embryos imaged from 34 hpf onwards.

H     Aperiodic $K_{OU}$ covariance model indicating differences in rate of correlation decay between subsequent peaks corresponding to examples in (F, G).

I     Example of one experiment representing quantification of noise by aperiodic lengthscale in one CTRL versus one MBSm embryo; bars indicate mean with 95% confidence intervals.

J     Pairwise comparison of aperiodic lengthscale; dots indicate median per experiment from CTRL (15 cells, 5 cells, 13 cells, 7 cells; 4 embryos) and MBSm (14 cells, 5 + 12 cells, 4 + 8 cells, 13 cells; 6 embryos); paired *t*-test with two-tailed significance for *$P < 0.05$.

state when it is de-repressed. This minimal network motif assumes only one further gene regulatory interaction, namely that gene X may self-activate (Fig 5A and Materials and Methods—model of downstream signal). We generated different hypothetical upstream dynamics by sampling traces from an OU Gaussian process characterized by different levels of high-frequency noise encoded by the $K_{OU}$ covariance function (Fig 2E). This allowed us to directly vary the aperiodic lengthscale $\alpha_{OU}$ of the gene Y. We found that the probability to switch towards the high X expression within a finite observation window decreased with increasing aperiodic lengthscale of Y. For slowly varying input Y (Fig 5B and C; case 1), the probability that the gene X is turned on is highest. For faster varying input (Fig 5B and C; case 2), the waiting time before the gene switches to high expression is increased, and in individual cases, the switch may not happen within the observation time window. For a quickly varying input (Fig 5B and C; case3), gene X does not become expressed. Thus, this network motif is highly sensitive to the aperiodic lengthscale in the dynamics of its repressing gene. Our mathematical model predicts that the loss of oscillatory expression and increased noise in the form of fast fluctuations in Her6 expression can impede the upregulation of downstream genes that mediate a cell state transition, such as pro-neural genes. However, we do know that *in vivo* Her6 is downregulated as cells progress towards differentiation. To explore a mechanism by which this may take place, we have introduced a negative feedback (direct or indirect, dashed line) from the target X towards the input Y (Fig 5D). Similar simulations as in the reduced model (Fig 5A) show that the input Y (Her6) is now also regulated, but the downregulation closely follows the cell state transition rather than initiate it (Fig 5D–F). At present, our computational model (Fig 5) is qualitative, rather than quantitative, and although it agrees with experimental data, a fully parameterized model could be developed further based on experimental evidence.

**Co-expression of *her6* with *elavl3* at the single cell level supports the model**

To test the model above, first, we characterized in more detail the expression of *her6* in relation to progenitor/differentiation markers. Triple-fluorescent whole-mount *in situ* hybridization (WM-ISH) staining was performed in embryos at 34 hpf (Fig 6A) including a progenitor marker, *gfap* and an early neuronal differentiation

marker, *elavl3,* which is switched off in more basally located mature neurons (Lyons *et al*, 2003). As expected, e*lavl3* and *gfap* are expressed in largely non-overlapping regions found in the apico-basal axis of the hindbrain (Fig 6A). However, a band of cells that co-express *gfap* and *elavl3* was identified, which we propose reflects a transitory state (Fig 6A—transition zone T), as *gfap*-expressing progenitors progress to *elav3*-expressing early differentiating neurons (Fig 6A, neurogenic zone N). The column of *her6* expression spans these domains, suggesting that it is expressed in progenitors (*gfap*(+)/*elavl3*(−)), early differentiating neurons (*gfap*(−)/*elavl3*(+)) and in cells of the identified transitory state (*gfap*(+)/*elavl3*(+)). *her6* was not detected in cells located more basally, suggesting that it is downregulated along the neuronal differentiation pathway and switched off in mature neurons (Fig 6A).

To determine whether *her6* and *elavl3* are co-expressed at the single cell level, we performed double smFISH for *her6* and *elavl3* and analysed single cells paying particular attention to bright spots in the nucleus, which denote the site of transcription. This analysis confirmed that progenitors express *her6* and differentiated neurons express *elavl3* (not shown), but most importantly, it also showed the existence of cells where the 2 genes are co-expressed (Fig 6B). This strongly suggests that, at single cell level, the onset of *elavl3* expression preceded the switching off of *her6* thus providing support for the computational model where a downstream gene X is activated independently or before a change in the level of gene Y (Her6; Fig 5). Elavl3 is a good candidate for fulfilling the role of downstream target X because ChiP-seq experiments have revealed Hes1 binding sites on *elavl3*'s regulatory region (Consortium, 2012, GEO reference numbers: GSM2825430 and GSM2422987 and preprint: Minchington *et al*, 2020), and reciprocally, evidence has been reported indicating indirect negative feedback (Coolen *et al*, 2012), which is consistent with the opposing functions of Her6/Elavl3. Future experiments will be needed to characterize the interactions of Her6 with downstream targets.

**Changes in Her6 dynamics/levels affect cell state progression *in vivo***

To determine whether the changes in dynamic expression of Her6 have phenotypic consequences, F0 MBSm embryos were generated (Materials and Methods—microinjection and genotyping) and investigated by triple-fluorescent WM-ISH. In transverse sections, this

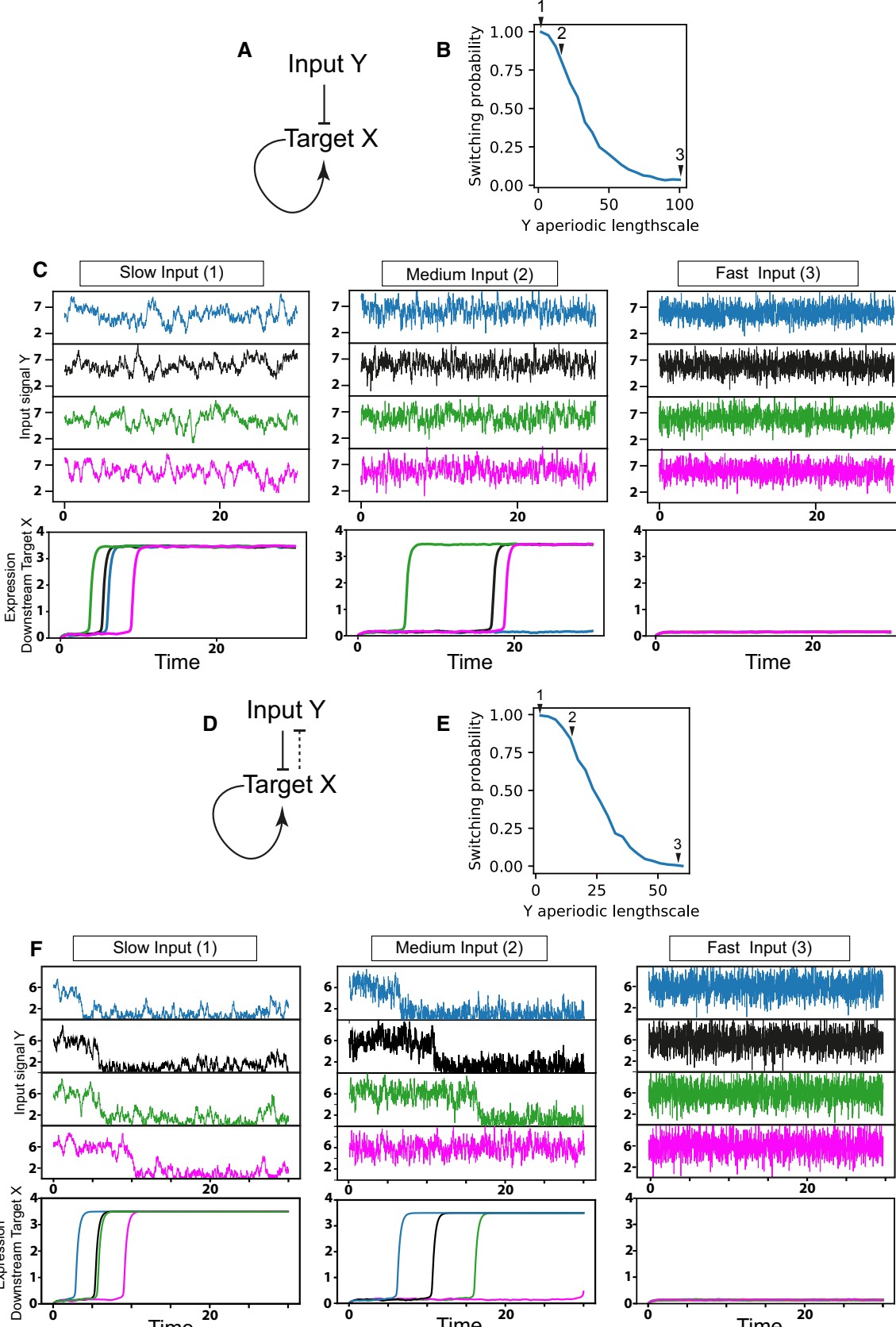

**Figure 5.**

**Figure 5.  Mathematical model exploring the effect of changes in Her6 dynamics on a downstream target.**

A   Network motif representing the interaction between a repressing gene Y (indicative of Her6) acting as input onto a downstream target gene X and self-activation of X.

B   Probability that the downstream target X switches to high expression from an initial off state; mathematical modelling shows that the probability decreases as the aperiodic lengthscale in the dynamics of Y increases.

C   Example of gene expression dynamics of Y and X for different scenarios corresponding to slow, medium and fast input, as quantified by aperiodic lengthscale ($\alpha_{OU}$) levels highlighted in b arrows, with $\alpha_{OU}$ = 2, 15 and 100, respectively; multiple stochastic examples are shown for each scenario (top), and matching dynamics of X and Y are presented (bottom) in corresponding colours between the two panels; for high-frequency input 3, X does not turn on within the observation window.

D   Extended network motif based on (A) and including an additional repressive interaction from X onto Y used to explore changes in fluctuations as well as Y levels (indicative of Her6 downregulation).

E   Probability that the downstream target X recapitulates switching from an initial off state to high expression similarly as in (B).

F   Example of gene expression dynamics observed for X and Y in the case of slow, medium and fast input scenarios corresponding to aperiodic lengthscale ($\alpha_{OU}$) levels marked in e arrows, with $\alpha_{OU}$ = 2, 15 and 60, respectively; multiple stochastic examples are shown for each scenario (top), and matching dynamics of X and Y are presented (bottom) in corresponding colours between the two panels; for high-frequency input 3, X does not turn on within the observation window.

showed a dorso-ventral expansion in the *gfap*(+)/*elavl3*(−) progenitors and the *gfap*(+)/*elavl3*(+) transitory progenitors in MBSm embryos compared to control at 52 hpf (Fig 6C and D bottom versus top panels). This was accompanied by a lack of the *gfap*(−)/*elavl3* (+), suggesting that early differentiating neurons were not present in MBSm (Fig 6C and D bottom panels). Quantitative analysis of *elavl3* and *gfap* levels in the Her6-expressing region encompassed T and N zones in control and T zone only in MBSm (Fig 6E–H), confirming the observed phenotype of reduced *elavl3* in an expanded double-positive T domain (Fig 6C and D, Appendix Fig S5A and B). We did not observe a phenotype at 28 hpf when comparing CTRL to MBSm embryos (Appendix Fig S5C–E); this is consistent with the miR-9 late expression and its modulation of neurogenesis in late hindbrain development.

We further tested the phenotype experimentally by investigating the effect of MBSm on other downstream targets that would normally be expressed when Her6 is downregulated. The late proneural transcription factor, basic helix-loop-helix gene, *neuroD4*, is known to be downstream and regulated by Her/Hes family members (Park *et al*, 2003; Bae *et al*, 2005); therefore, we looked at the expression of *neuroD4*, known as a marker of neuronal commitment. Consistent with the model prediction, we frequently observed a decrease in the expression of *neuroD4* in MBSm embryos (Fig 6I and J, Appendix Fig S5F).

Taken together, these findings suggest that in the absence of miR-9 regulation, neural progenitor cells do not progress through their normal cell state transition from progenitor to neuron and instead accumulate in a transitory state expressing both pluripotency and pro-neural markers.

## Discussion

Despite the intense interest in the biological role and consequences of protein expression noise (Eling *et al*, 2019), our understanding has been hampered by the lack of evidence of the dynamic properties of such noise. Indeed, with the exception of live study of transcriptional bursting, our current understanding of biological protein and gene expression noise is based on snapshot measurements of variability across populations, such as phenotypic or molecular variability of individuals and/or variability in molecular abundance between cells (often determined by scRNA seq), variability of fluorescence in single cells (reviewed in Eling *et al*, 2019) or simply the inappropriate re-expression of genes (Burgold *et al*, 2019).

Similarly, the effect of miRNAs in controlling noise has been based on artificial synthetic systems or analysis of static measurements (Li *et al*, 2009; Siciliano *et al*, 2013; Schmiedel *et al*, 2015). Because these methods are done at population level, measurements of noise have been limited to quantification of variability that can be measured by these methods, namely the standard deviation over the mean (coefficient of variation) and variance over the mean (noise strength or Fano factor) reviewed in Eling *et al* (2019), Kaern *et al* (2005). Here, we have used the power of Zebrafish as an experimental system that combines live imaging with experimental perturbation, to interrogate the molecular dynamics of cell state transitions during neural development. This allows us to characterize for the first time expression noise with fine time-resolution in order to understand its regulation and functional significance.

We have focused on Her6, a key transcriptional repressor that belongs to a family of Her/Hes genes that have been shown to oscillate during vertebrate somitogenesis and mammalian neural development, and are essential for these processes (Soroldoni & Oates, 2011; Delaune *et al*, 2012; Webb *et al*, 2016; Kageyama *et al*, 2019). Using an endogenous CRISPR-mediated knock-in fluorescent tag, we report for the first time the dynamics of a Her/Hes family member as they occur in real time, at the single cell level and in intact neural tissue. Our work was carried out in homozygous reporter Zebrafish; thus, the reported dynamics reflect what the cells experience. We found that Her6 expression undergoes a transition from aperiodic fluctuations ("noisy" expression) to oscillations with a dominant ultradian periodicity of 1–2 h as neurogenesis proceeds (Fig 7A). This is an example of beneficial use of noise in a biological system in driving oscillatory gene expression; by analogy to a concept in Engineering and Neuroscience where signal properties can be enhanced by noise, we suggest that this is a case of stochastic resonance or stochastic facilitation (reviewed in Paulsson *et al*, 2000; Hanggi, 2002; Moss *et al*, 2004; McDonnell & Abbott, 2009; McDonnell & Ward, 2011). Our observation of *in vivo* stochastic resonance is consistent with our recent experimentally informed (*ex vivo*) analysis that noise primes oscillatory expression during mouse neural development (Manning *et al*, 2019). It is also consistent with our previous computational modelling, which showed that increased stochasticity expands the parameter space where Hes1 oscillates (Phillips *et al*, 2016). Here, we were able to show that in Zebrafish the switch from noisy to oscillatory dynamics coincides temporally with the onset of miR-9 expression (Fig 7A), which has been previously proposed to target post-transcriptionally *hes1* and *her6* (Bonev *et al*, 2011, 2012; Coolen *et al*, 2012).

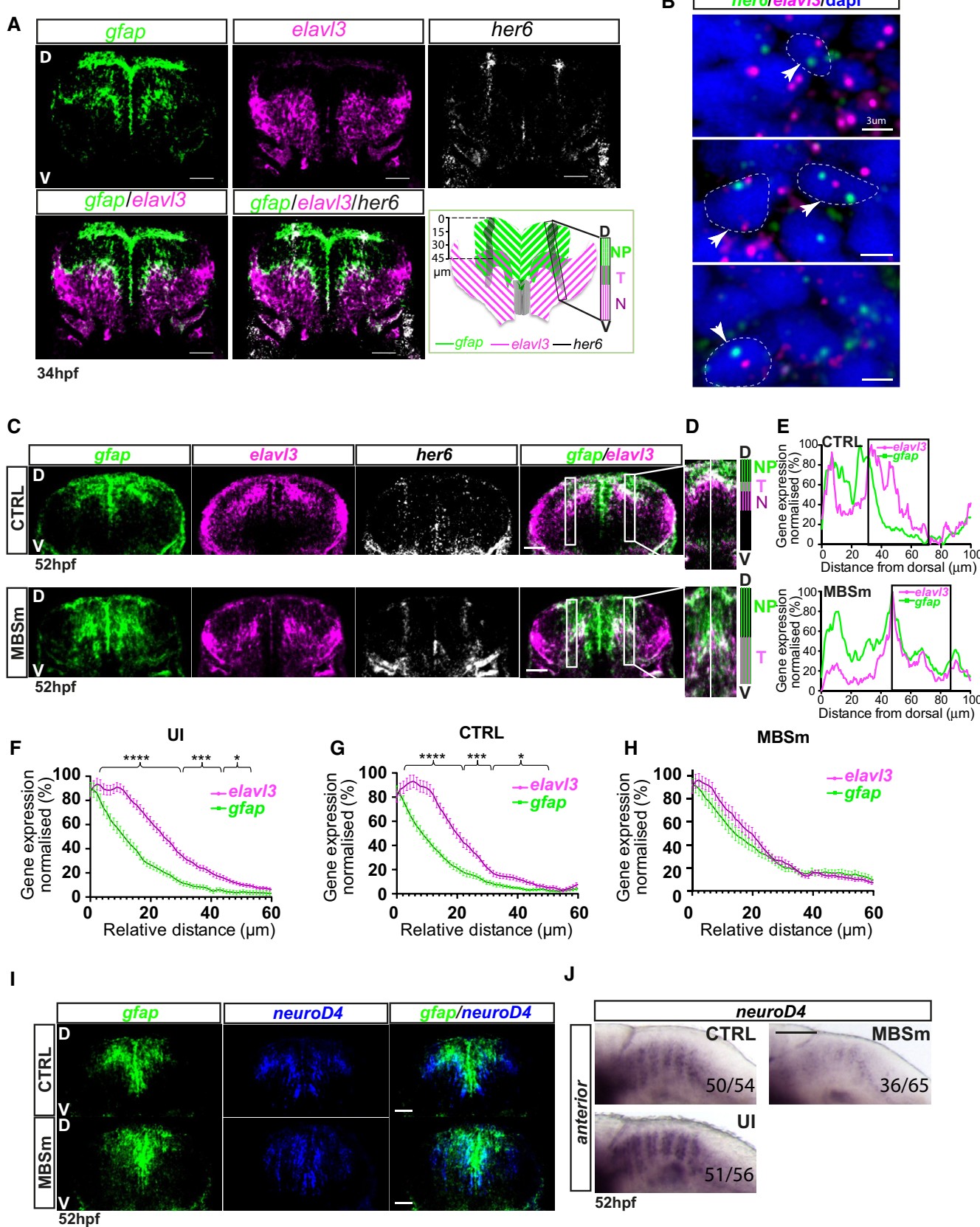

**Figure 6.**

**Figure 6.  Changes in cell fate decisions in the absence of miR-9 regulation.**

A   Representative examples of triple-fluorescent whole-mount WM-ISH labelling of *gfap* (green), *elavl3* (magenta) and *her6* (grey) domains of expression (top panels) in hindbrain rhombomere6 (r6) in wild-type embryo observed at 34 hpf; merged images indicate how the *her6* expression domain overlaps with the progenitor zone (NP = *gfap*(+)/*elavl3*(−)), transition zone (T = *gfap*(+)/*elavl3*(+)) and neurogenic zone (N = *gfap*(−)/*elavl3*(+)); bottom-right panel, schematic representation showing the *her6* domain spanning the NP, T and N zones with quantification of distances from dorsal and transversal view; annotations denote dorsal (D) and ventral (V); scale bar 30 μm.

B   Single-molecule fluorescent *in situ* hybridization (smFISH) showing *her6* (green), *elavl3* (magenta) and DAPI nuclear staining (blue) obtained from hindbrain (r6) sections of wild-type embryo at 34 hpf; head arrows indicate examples of co-existence of transcriptional active sites for *her6* and *elavl3*; scale bar 3 μm.

C   Triple-fluorescent WM-ISH for *gfap* (green) and *elavl3* (magenta) and *her6* (grey) in the hindbrain (r6) of CTRL (top) and MBSm (bottom) embryos at 52 hpf; dorsal (D); ventral (V); scale bar 30 μm.

D   Magnification of inset from merged *gfap*/*elavl3* in (c) showing CTRL (top) and MBSm (bottom) embryos with corresponding NP/T/N and NP/T zones, respectively; dorsal (D); ventral (V); scale bar 30 μm.

E   Normalized intensity mean of *elavl3* and *gfap* along the DV axis spanning the NP/T/N zones in CTRL and NP/T zones in MBSm, respectively; region of interest (ROI) delineates high *elavl3* versus *gfap* in CTRL (T and N zones), while in MBSm only the T zone is observed, overlap of *elavl3* and *gfap* intensity mean peaks.

F–H Normalized mean of *elavl3* and *gfap* intensities in ROI observed in: (f) uninjected (UI) (5 embryos, 62 slices), (G) control (CTRL) (5 embryos, 35 slices) and (h) MBSm (8 embryos, 62 slices) conditions; bars represent mean and SEM; multiple *t*-test with Benjamini, Krieger and Yekuteli discovery, significance: \*$P < 0.05$, \*\*\*$P < 0.001$, \*\*\*\*$P < 0.0001$.

I   Transverse sections of double-fluorescent WM-ISH for *gfap* (green) and *neuroD4* (blue) in the hindbrain (r6) of CTRL (top) and MBSm (bottom) embryos at 52 hpf; dorsal (D); ventral (V); scale bar 30 μm.

J   Chromogenic WM-ISH showing *neuroD4* expression intensity in uninjected (UI) 51/56 embryos (91%), CTRL 50/54 embryos (93%) and mutant (MBSm); 36/65 embryos (55%); longitudinal view, scale bar 100 μm.

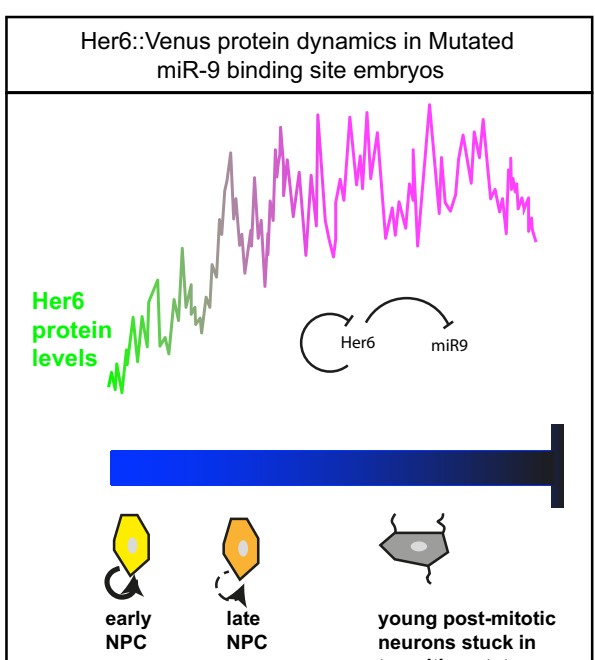

**Figure 7.  The role of miR-9 regulation on Her6 dynamic expression during hindbrain development.**

A  Her6 protein expression reveals a transition from irregular fluctuations (noisy expression) to oscillatory dynamics as differentiation proceeds; this coincides temporally with the onset of miR-9 expression.

B  Without the influence of miR-9, Her6 expression does not evolve away from the noisy into the oscillatory regime during development and is accompanied by a failure of the natural reduction of Her6 protein levels and impaired progression towards neural fate.

Using CRISPR-mediated knockdown of the miR-9 binding site in the *her6* 3′UTR, we functionally tested the role of miR-9 in modulating Her6 dynamics *in vivo*. An important finding of the present work is that miR-9 is necessary for the oscillatory behaviour of Her6 to

emerge (Fig 7B). Without the influence of miR-9, Her6 expression does not evolve away from the "noisy" into the oscillatory regime during development. In the absence of miR-9 influence, we identify an increase in noise frequency, measured by aperiodic lengthscale

of Her6 signal and a failure to downregulate Her6 to normal levels during development.

These findings prompted us to investigate the legitimate question of how the change in Her6 levels might relate to the change in Her6 dynamics and which one is likely to be most important. As there was no pre-existing framework on the impact of high lengthscale noise on gene expression, we developed a new computational model that allowed us to test its impact on downstream targets. We found that the activation of a downstream target does not require the downregulation of the upstream repressor (i.e. Her6) and that in principle, high-frequency noise of the repressor could be sufficient in preventing the activation of downstream targets. Our model further shows that under two reasonable assumptions, that is auto-activation of the target and a negative feedback from the target to Her6 (direct or indirect feedback), a cell state transition can occur that results in the downregulation of Her6 rather than the downregulation causing it. In this conceptual framework, the high-frequency noise would dominate Her6 function in the absence of miR-9 regulation and would secondarily cause a failure of Her6 downregulation.

This model is consistent with experimental results whereby we observed the occurrence of a deficiency to upregulate early neuronal genes (*neuroD4* and *elavl3*) and a concomitant failure of the natural reduction of Her6 protein levels during development in miR-9 binding mutants, suggesting that NPCs are unable to make a transition to differentiation. The phenotype is consistent with previous reduction of neurogenesis in miR-9 knockdown in Zebrafish (Coolen *et al*, 2012) and Xenopus (Bonev *et al*, 2011, 2012). A key experimental observation is that *elavl3* and *her6* are co-expressed at the transcriptional level in some neural progenitor cells, which are located in a "transitory" zone. This co-expression of a transcriptional repressor and a downstream target supports the concept that a cell state transition is initiated by a change in Her6 dynamics (for which miR-9 is important), while a downregulation of Her6 follows soon after and reinforces this transition. Full validation of this finding will require the identification of more markers that can be used to monitor entry and exit from the transitory "progenitor to neuronal" state as well as the simultaneous monitoring of Her6 protein dynamics with the transcriptional behaviour of early neuronal markers. Nevertheless, going beyond previous findings, our study suggests that the dynamic properties of noise, i.e. noise frequency, can impede Her6 oscillations and that these effects are alleviated by miR-9 (Fig 7). We propose that the molecular phenotype of miR-9 binding mutants is an example of (molecular) inverse stochastic resonance (Fig 7B) analogous to the term used in Engineering and Neuroscience to describe the inhibitory effect of noise on rhythmic neuronal firing (reviewed in Uzuntarla *et al*, 2013; Bacic *et al*, 2018).

The model we propose contrasts with the more traditional view of cell state transitions, which is based on protein expression levels crossing an activation/deactivation level before a subsequent fate change can occur. In such scenario, the role of the microRNA regulation would be solely to control the "relaxation" rate of the protein level (e.g. Cassidy *et al*, 2019). Our model does not exclude a role for protein level threshold and is indeed not mutually exclusive with this traditional view. In our model, the change in protein level is likely to be a consequence of the change in dynamics and it could act to reinforce the directionality of the state transition, effectively placing noise, oscillations and relaxation in a temporal, causative order. In this way, our work encourages the consideration of cell state transitions as a complex problem that integrates short-term and long-term protein dynamics. Furthermore, our model, with its emphasis on dynamics rather than level as the initiating event of cell state transitions, is well positioned to explain the intriguing and widespread observation of transient co-expression of opposing fate determinants in cells (e.g. Allison *et al*, 2018; Bergiers *et al*, 2018).

In conclusion, we have shown *in vivo* that neural progenitors express Her6 dynamically, undergoing a transition from noisy to oscillatory as neurogenesis proceeds (Fig 7A). While in cultured mouse cells, miR-9 can dampen the Hes1 oscillator (Bonev *et al*, 2012), a finding that has been computationally analysed (Tan *et al*, 2012; Goodfellow *et al*, 2014; Phillips *et al*, 2016), we showed here that *in vivo*, miR-9 has an important dynamic noise optimization role. In the absence of miR-9 regulation, progenitors undergo molecular inverse stochastic resonance, where the dynamic properties of noise impair Her6 oscillatory activity and cell state progression. Finally, our findings suggest that Her6 oscillations are necessary for progenitor's transition to differentiation to occur.

## Materials and Methods

### Research animals

Animal experiments were performed under UK Home Office project licences (PFDA14F2D) within the conditions of the Animal (Scientific Procedures) Act 1986. Animals were only handled by personal licence holders.

### Generation of *her6::Venus* knock-in line

We used CRISPR/Cas9 technology combined with a DNA donor to generate a reporter that would be suitable for live imaging. This involved an in-frame fusion of the fluorescent moiety, *Venus*, to the C-terminus of the endogenous *her6*, placed before the 3′UTR (Fig 1B and Materials and Methods for details on DNA donor design and guide RNA selection), anticipating that destabilization sequences within the Her6 protein would similarly destabilize the fusion protein. We first identified the single-guide RNA (sgRNA) to target *her6* exon 4, at the stop codon area using Addgene webpage, http://www.addgene.org/crispr/reference/#protocols. Two or more software packages were utilized to choose the top scored sgRNAs, based in high efficiency and low off-target effect (Materials and Methods—preparation of Cas9 and sgRNAs section). We experimentally tested the sgRNA by high-resolution melt (HRM) (see Materials and Methods—microinjection and genotyping) and chose the sgRNA with highest efficiency.

Further, we designed a DNA donor with either arms as big as 1 kb (Zu *et al*, 2013), the left arm (LA) contained *her6* exon2_intron2_exon3_intron3_exon4 and the right arm (RA) *her6* exon4, we destroyed the sgRNA target site by inserting *linker_Venus* within. In order to generate the reporter controlled by endogenous Her6 expression, we deleted the STOP codon from *her6* gene and the ATG codon from *Venus*. To avoid the inherent toxicity of linear DNA injection in fish, two CRISPR target sites flanked the DNA donor; thus, the injected DNA donor could get excised from the circular DNA (that contains DNA donor and vector) by the Cas9nls and sgRNA once injected in the embryo hence, providing the linear

template for the DNA repair (Irion *et al*, 2014) (see Materials and Methods—molecular cloning).

Next, we co-injected the DNA donor with the sgRNA and Cas9nls mRNA. Later, we identified the *her6::Venus* F0 adult fish carrying germline transmission (GLT) following the method described online (preparation of Cas9nls and sgRNA in Methods—microinjection and genotyping).

## Statistical testing

Comparative analysis between embryos at multiple conditions was carried out in GraphPad Prism 8.0. Quantitative data are presented per condition as box plot analysis with bars indicating median and interquartile range (5 and 95%) and statistically significant conditions reported for $P < 0.05$. Comparisons between normal distributed conditions were performed using one-way ANOVA with Bonferroni multiple comparisons test. Protein abundance from fluorescence correlation spectroscopy was analysed using Kruskal–Wallis with Dunn's multiple comparison correction test. Dynamic parameters at multiple stages were presented as violin plots (bars: median and interquartile range; violin shape: distribution) and compared using a Mann–Whitney two-tailed test. Dynamic parameters were compared between control and mutant embryos as paired median values per experiment using a paired *t*-test with two-tailed significance. Linear correlations with Z position were tested using Pearson's correlation coefficient. Non-linear correlations between signals collected from the same cells were tested using Spearman's rank correlation coefficient. Intensity levels observed across the DV axis were adjusted to the same scale by linear interpolation and compared between conditions using multiple *t*-tests with Benjamini, Krieger and Yekuteli discovery, $Q = 1$.

## Code and data availability

Detection of oscillators, aperiodic lengthscale and frequency analysis (Materials and Methods—dynamic data analysis and frequency analysis) were performed using custom MATLAB routines that will be deposited at https://github.com/VBiga/her6noise. The model of the network motif (Materials and Methods—model of downstream signal) was implemented in Python, and code is found at https://github.com/kursawe/hesdynamics. Single cell Her6::Venus, mKeima-H2B raw and processed intensity traces are available upon request from the corresponding authors.

## Molecular cloning

The DNA donor was constructed in the pCRII vector (Life Technologies). The *her6*-LA, *her6*-RA and linker-*Venus* were generated by PCR, using genomic DNA of wild-type (AB) fish and *her1:her1-linker-Venus* (Delaune *et al*, 2012), respectively, using specific primers (see Appendix Table S1). Primers 1 and 4 contained the sgRNA target site and primer 5 did not include the start coding sequence for *Venus*. The PCR products were cloned into pCRII vector followed by sequential subcloning steps to assemble the DNA donor: First, we used SalI/SpeI to subclone LA into pCRII_linker-Venus, and then, we used KpnI to subclone linker-Venus_LA into pCRII _RA. The pCS2 + mKeima-H2B was generated by sequential PCR, restriction enzyme treatment and subcloning steps, using the plasmid mKeima-Red-N1 (Addgene #54597) as template. The

pCS2 + HA::Her6 and pCS2 + Her6::Venus were generated by sequential PCR and subcloning steps, using as a template cDNA obtained by reverse transcription (RT) from embryonic mRNA (see Appendix Table S1 for respective primer set).

## Preparation of Cas9nls and sgRNAs

The Cas9nls mRNA was generated from pT3TS-nls-zCas9-nls plasmid obtained from Addgene #46757 following the protocol described by Li-En Jao (Jao *et al*, 2013). The Cas9nls protein was obtained from New England Biolabs M0641M. The sgRNA target sites were identified using the CRISPRdirect (http://crispr.dbcls.jp/) and Target Finder (Feng Zhang lab http://crispr.mit.edu/).

To synthetize the sgRNA used to generate the *her6::Venus* knock-in, selected oligonucleotides (Appendix Table S1; oligos 7 and 8) were annealed and cloned into pT7-gRNA plasmid #46759 (Addgene), following the protocol described by Jao *et al* (2013). The correct clones were linearized with BamHI, and transcription of sgRNA was carried out using MEGAshortscript T7 kit (Ambion/ Invitrogen) with 100–400 ng of purified linearized DNA following the manufacturer's instructions. The sgRNA was purified using MEGAclear™ Transcription Clean-Up Kit.

For Her6 miR-9 binding site (MBS) mutation, sgRNAs were generated following CRISPRscan protocol (Moreno-Mateos *et al*, 2015) using the oligonucleotides described in Appendix Table S1, oligos 25 or 26. The PCR fragments were purified using Qiagen columns, and transcriptions were carried out as described above.

## Microinjection and genotyping

To generate the *her6::Venus* knock-in (Ki), one-cell stage wild-type AB Zebrafish embryos were injected with < 1 nl of a solution containing 150 ng/µl Cas9nls mRNA, 200 ng/µl sgRNA and 20 ng/µl circular DNA donor in 0.05% phenol red. To generate MBS mutation, one-cell stage *her6::Venus* Ki embryos were injected with < 1 nl of a solution containing 185 ng/µl Cas9nls protein, 125 ng/µl sgRNA, 40 ng/µl caax-mRFP mRNA and 40 ng/µl mKeima-H2B mRNA in 0.05% phenol red. Embryos were injected with minimal amounts of MBS sgRNA (F0) to not have overt phenotype at the macroscopic level during the experimental period (24–52 hpf), thus minimizing the chances of non-specific toxicity.

To evaluate whether the sgRNA was generating mutation, genomic DNA was extracted from 3 to 4dpf embryos using 50 µl NP lysis buffer per embryo (10 mM Tris pH 8, 1 mM EDTA, 80 mM KCl, 0.3% NP-40 and 0.3% Tween) and 0.5 µg/µl Proteinase K (Roche) for 3–4 h at 55°C, 15 min at 95°C and then stored at 4°C. Then, high-resolution melt (HRM) was performed using Melt Doc kit following manufacturer instructions, and specific primer set was used according *her6::Venus* Ki or MBS mutation (see Appendix Table S1; oligo 9/10 or 27/28, respectively). Further, to evaluate efficiency of the sgRNA, PCR was performed per embryo (using the same primers for HRM) and the amplicon obtained was cloned into pCRII vector and transformed into bacteria Top10. Then, 8 bacterial colonies per embryo were miniprep and sequenced. The efficiency per cent was calculated per embryo according to the number of sequences mutated in the total of 8 sequences per embryo.

To assess homology direct repair (HDR) in F0 or F1 progeny, genomic extraction was performed as described above followed by

PCR and agarose gel (Appendix Table S1). The PCR products were cloned into pCRII vector and sequenced. It is relevant to mention that the repair of the Her6-RA was not perfect as it included part of the vector that was used to generate the DNA donor; nevertheless, we expected this insertion to not affect the protein expression, and dynamics, as the Her6-RA included the 3′UTR and Her6 polyadenylation site.

To identify F1 progeny with germline transmission (GLT), 3–5dpf embryos were fin clipped following the protocol described by Robert Wilkinson (Wilkinson *et al*, 2013) with modifications. Sylgard (Sigma, Cat # 761028)-coated 10-cm dish was prepared for dissections. Embryos were placed into Sylgard-coated dish containing E3 medium with 0.1% Tricaine (Sigma, UK) and 2% BSA (Sigma, UK). Once clipped the fin, the embryo was transferred to E3 medium and the biopsy was transferred to PCR tube for genomic extraction. Genomic extraction was carried out in 10 μl volume using Phire Animal Tissue Direct PCR kit (Thermo Scientific, Cat # F-140WH). 2 μl of the supernatant was used for 10 μl qPCR using primers 9 and 12. The derivative of the melting curve was used to identify the positive GLT fish (Appendix Fig S1H).

The frequency of repaired DNA with the insertion of *venus* in F0 fish was 30% with a decrease to 3.3% (2/60) from those F0 adults carrying germline transmission (GLT), identified by the PCR fragment of the expected size (Appendix Figs S1E and S2C). Sequencing of F1 embryos throughout the endogenous-donor DNA fusion was used to ensure that the reading frame was correctly maintained (Appendix Fig S2D). The F1 embryos were selected for the presence of the reporter by fin clipping at 3dpf and genotyped by qPCR making use of the derivative of the melting curve (Appendix Fig S1F–H). The positive fish were confirmed by fin clipping at 8wpf and visualization of the right size amplicon (Appendix Fig S1D and I and Materials and Methods—microinjection and genotyping for strategy to obtain F1 generation).

## Whole-mount chromogenic and fluorescence *in situ* hybridization and sectioning

Chromogenic *in situ* hybridization was carried out as described by Christine Thisse (Thisse & Thisse, 2008). Multicolour fluorescence *in situ* hybridization was developed using tyramide amplification after addition of probes and antibodies conjugated to horseradish peroxidase (Lea *et al*, 2012). RNA probes for Elavl3, GFAP, Neurod4, Her6, Venus and pri-miR-9-4 were PCR amplified and cloned into pCRII vector using the primers in Appendix Table S1. Eplin probe (to stain somites and faithfully count them at 72 hpf) was generated from a plasmid kindly gifted by Andrew Oates. miR-9 LNA 3′5′Dig probe was purchased from Exiqon.

Sections were obtained as described in Dubaissi (Dubaissi *et al*, 2012) with modifications. Embryos were embedded in 25% fish gelatine for a minimum of 24 h. 18-μm-thick sections were collected and transferred onto superfrost glass slides. The slides were air-dried for 6 h under fume hood and washed for 2 min in PBS only before mounting.

## Protein Half-life

One-cell stage wild-type embryos were injected with 80 pg of Her6:: Venus or HA::Her6 mRNA. Protein half-life was performed using 200 μM cycloheximide and incubation started at 128–256 cells, development stage. Pools of 20 embryos were collected at 0, 5, 10, 15 and 20 min. Samples were lysed with 2 μl/emb of Ginzburg Fish Ringer lysis buffer (110 mM NaCl, 3.35 mM KCl, 25 mM CaCl$_2$ and 2.4 mM NaHCO3) and washed twice with 500 μl of Ginzburg Fish Ringer wash buffer (110 mM NaCl, 3.5 mM KCl, 2.7 mM CaCl$_2$ and 10 mM Tris pH 8.8). The pellet was resuspended in 2 μl/emb of 1× Laemmli buffer. Western blots were performed using 4–20% Tris-glycine acrylamide gels (NuSep) and Trans-Blot Turbo Midi Nitrocellulose Transfer Pack (Bio-Rad) and developed with Pierce ECL substrate (Thermo Fisher Scientific). Antibodies used were anti-GFP (mouse Roche 1181446001), anti-HA-HRP (rat monoclonal Roche 2013819) and anti-alpha-tubulin (clone DM1A Sigma T9026).

## Fluorescent correlation spectroscopy (FCS)

For FCS experiments, embryos were mounted in a customized metallic device, with microscope slide shape but hollowed in the middle in order to fix a cover slip on either side. The embryos were mounted in between two cover slips with 1% low-melting agarose (Sigma) with the region of interest close to the cover slip. Snapshot images were collected with Zeiss LSM880 microscope with a C-Apochromat 40× 1.2 NA water objective. FCS signals were collected inside single nuclei in dorsal region of the hindbrain in intact embryo. Venus (EYFP) fluorescence was excited with 514 nm laser light and emission collected between 517 and 570 nm.

Data from individual cell nuclei were collected using 5 × 5 s runs at 0.15–0.3% laser power, which gave < 10% bleaching and a suitable count rate ~1 kHZ counts per molecule (CPM). To obtain molecule number, autocorrelation curves were fit to a two-component diffusion model with triplet state using an optimization Toolbox based on the Levenberg-Marquardt algorithm with initial conditions assuming a "fast" diffusion component 10× faster than the "slow" component as described in Smyllie *et al* (2016).

Measurements collected from cells exhibiting large spikes/drops in count rate or with low CPM (< 0.5 kHz), high triplet state (> 50%) or high bleaching (> 10%) were excluded from the final results. Number and brightness analysis of the count rate showed a high correlation with molecule number obtained from autocorrelation curve fitting. The effective confocal volume (CV) had been previously determined with mean 0.57 fl ± 0.11 fl (Bagnall *et al*, 2015) allowing conversion from molecule number to concentration. Single cell data of absolute protein number in the cell nucleus were obtained by adjusting concentration in CV to the average volumetric ratio between nuclear volume and confocal volume. Cell volumes were larger at 28 hpf (379.32 ± 26.39 fl), compared to 34 hpf (118.13 ± 5.9 fl). CTRL and MBSm at 34 hpf showed no significant differences in volume (volumetric ratio CTRL versus MBSm = 1.01).

## smFISH probe design and synthesis

The smFISH probes were designed using the probe design tool at http://www.biosearchtech.com/stellarisdesigner/. Depending on the GC content of the input sequence, the software can return varied size of probes, 18 and 22 nt, hence giving the largest number of probes at the maximum masking level. It also uses genome information for the given organism to avoid probes with potential off-target binding sites. Using the respective gene mature mRNA sequence, we

designed 29 probes for *her6* and 27 probes for *elavl3* (Appendix Table S2). The designed probes were synthesized and labelled with Quasar 570 (Cy3 replacement) for *her6* or Quasar 670 (Cy5 replacement) for *elavl3* at the 3′ ends at Life Technologies or Biosearch Technologies.

### Whole-mount smFISH

Whole-mount smFISH protocol for zebrafish embryos was developed by adapting smiFISH protocol from Marra *et al* (2019). Embryos were fixed in 4% formaldehyde in 1× PBS. After smFISH staining, embryos were embedded in 25% fish gelatin for a minimum of 24 h. 18-μm-thick sections were collected and transferred onto superfrost glass slides (VWR 631-0448). Slides were kept for maximum 24 h at −80°C and then were air-dried for 6 h under fume hood. Slides were washed for 2 min in PBS and mounted using Prolong Diamond Antifade Mountant with DAPI (Thermo Fisher P36962).

### smFISH microscopy and deconvolution

smFISH images were collected with Leica TCS SP8-inverted confocal microscope using objective 100×/1.4oil. We acquired three-dimensional stacks 512 × 512 pixels and z size 0.3 μm. The voxel size was $0.23 \times 0.23 \times 0.3$ μm. Quasar 570 and 670 were imaged with pinhole 1 AiryUnit and DAPI with pinhole 2 AiryUnit. Channels were sequentially imaged.

Deconvolution on confocal images was performed using Huygens Professional Software. As pre-processing steps, the images were adjusted for the "microscopic parameters" and for additional restoration such as "object stabilizer"; the latter was used to adjust for any drift during imaging. Following, we used the deconvolution Wizard tool, the two main factors to adjust during deconvolution were the background values and the signal-to-noise ratio. Background was manually measured for every image and channel, while the optimal signal-to-noise ratio identified for the images was value 3. After deconvolution, the images were generated with Imaris 9.3.

### Live imaging of whole developing hindbrain and image analysis

To study the overall Her6 expression pattern during hindbrain development, the embryos were laterally mounted in 1% low-melting agarose on glass-bottom dishes. Embryos were imaged using either Leica TCS SP5 upright confocal or Zeiss LSM 880 fast Airyscan microscopes.

For short imaging period, 20–40 hpf, we collected Her6::Venus images every 10 min using Leica TCS SP5 upright confocal microscope. Parameters used were similar to imaging for single cell tracking (see Material and Methods—live imaging for single cell tracking) with small modifications such as ×1 zoom, image size x:516.19 μm, y:516.19 μm and z:75–80 μm. For long imaging period, 30–70 hpf, and to include parallel imaging CTRL versus MBSm, we collected Her6::Venus images every 6 h using Zeiss LSM 880 fast Airyscan. Parameters used were similar to imaging for single cell tracking with small modifications such as ×1.2 zoom, image size x: 351.56 μm, y: 351.56 μm and z: 155.10 μm.

Using FIJI software, the images were 2D Max projection over time. Further, the borders of each rhombomere were drawn manually using freehand ROI (region of interest) creating function (imfreehand) over

time. This was possible due to the morphological characteristics of rhombomere boundaries. The ROI manager tool allowed us to obtain the intensity mean and area of each rhombomere over time. The graphs represent intensity mean over the area.

### Live imaging for single cell tracking

For single cell tracking, live embryos were mounted with forebrain facing down (Fig 2A) in 1% low-melting agarose on glass-bottom dishes (MatTek Corporation P50G-1.5-14-F), in order to collect a transversal view of the rhombomere 6. During the imaging period, the mounted embryos were maintained at 28°C using an in-line solution heater and heated stage, both controlled by a dual-channel heater controller (Warner Instruments), and the media was supplemented with 0.0045% 1-phenyl-2-thiourea and 0.1% tricaine. Images were collected as short as every 6 min to avoid sample bleaching.

Embryos were imaged using either Leica TCS SP5 upright confocal, TCS SP8 upright confocal or Zeiss LSM 880 fast Airyscan.

For Leica TCS SP5 and TCS SP8, we used HCX IRAPO 25 × 0.95 water dipping objective with 3.5× zoom, image size x:126.78–177.49 μm, y:63.39–88.74 μm, z:30–37.5 μm, Z size: 2–2.5 μm and pinhole 111.69 μm (2 AU). Channels were sequentially imaged with bidirectional scanning.

For Zeiss LSM 880 fast Airyscan, we used W Plan-Apochromat 20×/1.0 objective with 3× zoom, Image size x:132.79 μm, y:139.03 μm, z:37 ± 1 μm, Z size: 0.550 μm and pinhole 525 μm. Channels were sequentially imaged, and filters used were Track1: BP 420-480 + LP 605, Track2: BP420-480 + BP495-550 and Track 3: BP 420-480 + LP 605. Lasers used were Track1 561 nm: 2%, Track2 514 nm: 10% and Track3 458 nm: 10%.

### Single cell tracking

Due to high density of neural progenitor cells in the Her6-expressing area of the hindbrain, tracking from the mKeima-H2B (nuclear signal found in all cells) was not tractable. Instead, tracking was performed on a channel based on Her6::Venus that has been pre-processed to enhance nuclear separation. Specifically, we used the arithmetic tool in Imaris to produce a channel dedicated to tracking and we refer to it as the Her6::Venus segmented channel. Generating this involved two arithmetic operations. First, the mKeima-H2B channel (nuclei signal, Ch1) was subtracted from the mRFP-caax channel (membrane signal, Ch2) to remove spurious nuclear signal (background) and autofluorescence signal coming from apoptotic cells. Thus, a new channel referred to as caax-mRFP subtracted (Ch4) was generated representing an enhanced version of the surface marker. The second arithmetic operation involved subtracting the newly produced Ch4 from the Her6::Venus channel (Ch3) to better define the contour of cells expressing Her6, thus generating a new Her6::Venus-segmented channel (Ch5). An example output of these operations is included in Appendix Fig S3A.

Single NPCs found in rhombomere 6 were tracked in Imaris on the Her6::Venus-segmented channel (Ch5). This involved a combination of automated and manually curated steps.

As first step, nuclear detection was performed using automated "Spots tool", which can identify individual nuclear regions based on intensity similarity, followed by "Track over time" of the 3D object by using the Brownian motion algorithm. This produced series of

objects over the time, with one 3D bounded object per nucleus per timepoint that we referred to as a 3D spot (sphere). The diameter of the 3D spot was set to 5 μm, which is consistent with nuclear diameters measured from mKeima-H2B (approx. 6.5 μm). Thus, the 3D spot covers over 75% of the nuclear volume. Automated tracking very frequently produced errors including partially tracked nuclei and nuclei incorrectly assigned to a neighbouring track. To account for these technical problems, we manually curated the tracking data by merging/splitting of tracks and re-assigning nuclei to the correct track at each timepoint.

Additional tracks were generated by using the "Spots tool" in manual mode allowing us to manually place the sphere in the centre of the mass over time. We tested the accuracy of manual positioning by comparing centre of mass co-ordinates against automated spot tracking (Appendix Fig S3B). Although we could only do this for a minority of automated tracks of sufficient length (133 timepoints, $n = 3$), we found that the positional error was on average less than 0.5um and not affected by the length of the manual tracking. For data analysis, we collected Intensity mean from the raw Her6::Venus channel to provide a measure of concentration over time. In addition, we collected concentration of mKeima-H2B in the same cells as a measure of biological noise.

### Background fluorescence

We collected information on technical noise (comprising of autofluorescence and detector noise) by generating manual 3D tracks in areas of the sample that does not contain the fluorophore, i.e. a tissue area not expressing Her6 protein and an area outside of the tissue not expressing either Her6 or H2B; we refer to these as Venus background and H2B background, respectively. We collected a minimum of 2 backgrounds per experiment per channel representing a minimum of 4 background traces per embryo.

### Timeseries data pre-processing

Raw data timeseries for Her6 and nuclear marker H2B in the same cells were exported from Imaris tracks (see Single Cell Tracking) as intensity mean of Venus and mKeima over time, respectively. To account for photobleaching in each channel independently, we calculated the linear decay per channel and used this to linearly adjust the trend of Her6::Venus and mKeima-H2B over time in each experiment. As expected, the photobleaching of Venus was minimal, while mKeima was more pronounced in some experiments (examples in Appendix Fig S3C).

We noted a weak correlation between Her6::Venus and mKeima-H2B (Appendix Fig S3D), and we normalized the Venus signal to mKeima in order to remove fluctuations due to global concentration changes not specific to Her6 regulation. Finally, we removed the effect of long-term trends (above 3 h) in the Her6/H2B normalized signal as previously described in Phillips *et al* (2017) to generate the detrended Her6/H2B signal (examples included in Figs EV1 and EV4).

As expected from imaging through tissue, there was a small negative correlation of Her6::Venus and mKeima-H2B intensity associated with Z position, $r = -0.114$ and $r = -0.148$, respectively, when all nuclear intensities and time-points were plotted per channel (Appendix Fig S3E and F). However, the range of Z positions in a single cell over a 10- to 12-h track was rarely > 25 μm, and

therefore, the effects of Z position on Her6::Venus were found negligible and no correction for z was applied.

### Dynamic data analysis

The dynamic data analysis was performed using custom MATLAB routines amended from the approach in Phillips *et al* (2017) and code generated by Nick Phillips (deposited at https://github.com/ManchesterBioinference/GPosc) and using the GPML toolbox (details at http://gaussianprocess.org/gpml/code/matlab/doc/) (Rasmussen & Nickisch, 2010). Specifically, we use Gaussian processes that can represent timeseries in terms of a *mean function*, capturing changes in level over time and *a covariance function* describing fluctuations around the mean. The mean functions are estimated during detrending (see Timeseries data pre-processing). Periodic and aperiodic dynamic activity is characterized by two competing covariance functions based on Ornstein-Uhlenbeck (OU) (Gardiner, 2009), one encoding *aperiodic and stochastic* fluctuations, $K_{OU}(\tau) = \sigma \exp(-\alpha_{OU}\tau)$ and another encoding *periodic and stochastic* activity through an additional cos wave term, $K_{OUosc}(\tau) = \sigma \exp(-\alpha_{OUosc}\tau) \cdot \cos(\beta\tau)$. The signal variance, $\sigma$, is included in both models and is related to amplitude of the detrended Her6/H2B signal. Since we discuss amplitude separately in relation to level (see coefficient of variation), data are z-scored prior to analysis leading to $\sigma = 1$.

Stochasticity is described in both models by an exponential decay in correlation between subsequent peaks and parameterized by the rate of decay parameter, $\alpha_{OU}$, aperiodic lengthscale and, $\alpha_{OUosc}$, periodic lengthscale for $K_{OU}$ and $K_{OUosc}$ respectively. In addition, the periodic covariance model includes a cos wave term with frequency $\beta$ and related to ultradian period by $P = 2\pi/\beta$. Parameters are inferred from data by maximum likelihood techniques where likelihood represents the probability of the observed data under the model and includes a technical noise term.

#### *Calibration of technical noise from background*

We used experimental data to calibrate technical noise in each experiment, and this approach was first applied in (Phillips *et al*, 2017). Specifically, for the Venus/H2B measure we determined the variance of technical noise from timeseries collected from an area of the tissue not expressing Her6 but having expression of H2B (see Background fluorescence). For the H2B signal, we computed the variance of detector noise from an area of the sample not expressing H2B. Technical noise variance is then imposed during parameter estimation such that the residual signal representing the difference between the detrended data and the model fit is a random variable of the same variance as the background fluctuations. This calibration procedure ensures that the covariance models do not fit spurious fluctuations observed below the detection limit.

#### *Parameter estimation*

We have previously shown that fluorescence data acquired from tissue have a characteristic low signal-to-noise ratio (90%), which impacts log-likelihood, and thus, the estimation of multiple parameters required additional considerations for the detection of periodicity (Manning *et al*, 2019). To account for these problems, we defined a prior for the estimation of periodic lengthscale using the SmoothBox1 function in GPML defined as $SB1(\alpha) = S(\eta(\alpha–l)(1–S(\eta(\alpha–L)))$ where $S(z) = 1/1 + \exp(–z)$ with parameters $l = 1$, $L = 2$,

$\eta = 5$. In this way, we constrain the values of periodic lengthscale estimated under the OUosc model and improve the ability to discriminate between oscillatory and non-oscillatory cells.

### False discovery rate

We classified cells into oscillatory and non-oscillatory using the false discovery rate (FDR) approach detailed in Phillips *et al* (2017). Specifically, we used the log-likelihood ratio (LLR), a statistic that compares the likelihood of the periodic and aperiodic models such that a large LLR value indicates high probability of the timeseries being oscillatory while a low value of LLR indicates high probability of aperiodic fluctuations (non-oscillatory). Our approach uses synthetic aperiodic data from the estimated $K_{OU}$ model to produce LLR statistics for the null hypothesis. Using distributions of LLR from data and from the null hypothesis, we impose a 3% FDR threshold to determine statistically significant oscillatory activity. The FDR classification is performed independently per experiment. We compared the classification error of our approach by analysing H2B time series in the same cells and found that the rate of false positives was on average 4.5% ($n = 4$), which falls within the < 5% FDR and is similar to the desired value.

### Frequency analysis

Power spectrum analysis was implemented in MATLAB R2019a. Specifically, we reconstructed the power spectrum from single cell timeseries using the *periodogram.m* routine, which is a non-parametric technique using the Fourier transform to deconstruct the contribution of each frequency to the overall signal variance. Typically, this means that the spectrum $S(f)$ is computed at frequencies: $f_k = kF_s/N, k = \overline{0, N-1}$:

$$S(f) = \frac{1}{TF_s} \left| \sum_{n=0}^{T-1} x_T(n) e^{-j2\pi f_k\, n/F_s} \right|^2$$

where $x_T$ is a signal of finite length T acquired with $F_s$ sampling frequency.

We analysed the power spectrum of individual timeseries from detrended Venus, H2B and Venus/H2B signals for each dataset and used a fixed frequency range. Shorter length timeseries corresponding to Figs 2 and EV1 and EV2 were analysed using a Hamming window to produce a smoother appearance; meanwhile, a rectangular window was sufficient for analysis included in Figs EV3 and EV4. The window function is multiplied onto the signal in the time domain, $x_T(n) = x_{\text{detrended}}(n)w(n)$, and thus corresponds to kernel convolution in the frequency domain. The use of a window function is a standard way to improve the appearance of the power spectrum, and different options are discussed for example in (Smith, 1997).

We generated aggregate power spectra by averaging spectra over all cells per condition, and examples are included in Figs EV2A and EV4A. To allow a better visualization of peaks in the single cell power spectra, we used *periodogram.m* with the option "*psd*" whereby power is normalized by signal variance and examples are included in Figs EV2B and EV4B.

### Coherence from aggregate spectra

We quantified the occurrence of oscillatory activity in the Venus and Venus/H2B aggregate power spectra and compared this to H2B

power spectra using coherence (Alonso *et al*, 2007; Phillips *et al*, 2016). This measure expresses the concentration of power (as a percentage of total power) around a specific frequency typically corresponding to maximum power. To account for the fact that power distributions can show not only a single dominant peak but a range of frequencies where power is high, we used a polynomial fit (order 6) to smooth the shape of the power spectrum prior to identifying the peak frequency value corresponding to the maximum in the fitted power. Coherence was then calculated from the true power distribution over a set frequency interval corresponding to 10% of peak frequency and centred at the peak frequency value. Finally, we note that although increased values for coherence are indicative of the prevalence of oscillators its use is limited since it is a qualitative population measure.

### High-frequency content from single cell spectra

The aggregate power spectra in all conditions showed that power tapers off after frequency 1.5 (1/h) corresponding to 40 min but continues to show activity above the detection limit observed in the background (Figs EV2A and B, and EV4A and B). We chose this frequency value as a cut-off between the ultradian frequency (low-frequency) and the high-frequency range. From the single cell power spectral densities (Fig EV4D and E), we quantified the % contribution of high frequency in the Venus/H2B and H2B signals by integrating power across the high-frequency region and dividing by the area under the curve (Fig EV4F and G).

## Coefficient of Variation

Signal variability around the mean was analysed from the Her6:: Venus/mKeima-H2B single cell timeseries by using coefficient of variation (COV), a statistic denoting the ratio between standard deviation and the sample mean, reviewed in Eling *et al* (2019), Kaern *et al* (2005) and typically used to measure gene expression noise. We accounted for changes in mean level over time by using a local time window set to 1.5 h, and we refer to this window method as local COV (LCOV). In the case of pairwise CTRL and MBSm experiments, we calculated LCOV for every cell in CTRL and MBSm, obtained the median CTRL and MBSm values, and we report the LCOV ratio representing median LCOV in MBSm divided by the median LCOV in CTRL.

## Model of downstream signal response

We implemented a mathematical model for the response of the expression of a target gene $X$ to different dynamics of Her6 expression $Y$. A key component of the model is the simulation of Her6 dynamics with adjustable aperiodic lengthscale $\alpha_{OU}$.

### Simulation of Her6 fluctuations

In Fig 5A–C, we generated *in silico* Her6 dynamics by sampling traces from an OU process, which is introduced using the covariance function $K_{OU}$ in the main text. Varying the aperiodic lengthscale of the process can generate traces with different timescales of fluctuation, similar to the differences we see between the data of the MBS and the control experiments. In order to ensure non-negative values for these *in silico* traces, we added the Ornstein-Uhlenbeck traces to an otherwise constant and non-negative signal. Specifically, *in silico* Her6 dynamics in Fig 5B and C were generated by combining a

constant expression at a level of 6 arbitrary units with a sample from an OU process for which we used a variance parameter of 1.7 and different values for $\alpha_{OU}$ (case 1: $\alpha_{OU} = 2$, case 2: $\alpha_{OU} = 15$, case 3: $\alpha_{OU} = 100$). Once we have defined *in silico* Her6 dynamics $Y(t)$ in this way, the downstream response is calculated with a separate differential equation, outlined in the section "simulation of downstream signal response" below.

In Fig 5D–F, we allow for repression of Her6 ($Y$) by its downstream target $X$ by simulating $Y$ dynamics using the stochastic differential equation

$$\frac{dY}{dt} = \alpha_{OU} Y_{in} G_3(X(t)) - \alpha_{OU} Y + \sqrt{2\alpha_{OU}\sigma^2}\xi, \tag{1}$$

where $\alpha_{OU}$ is the aperiodic lengthscale, $Y_{in}$ is the initial level of Her6 expression, $\sigma^2$ is the variance parameter, and $X(t)$ is the expression level of the downstream target at time $t$. The repression function $G_3$ takes the form

$$G_3(X) = \frac{1}{1 + \left(\frac{X}{X_{0,Y}}\right)^n}.$$

This function takes the value 1 if the number of $X$ molecules is much smaller than the repression threshold $X_{0,Y}$, and 0 if the number of $X$ molecules is much larger than this threshold. The steepness of the transition is regulated by the Hill coefficient $n$.

Further, $\xi$ denotes $\delta$-correlated Gaussian white noise

$$\langle \xi(t_1)\xi(t_2) \rangle = \delta(t_1 - t_2).$$

In the limit of low expression levels of $X$, this stochastic differential equation describes an OU process with aperiodic lengthscale $\alpha_{OU}$, variance $\sigma^2$ and mean expression $Y_{in}$. When $X$ expression levels increase, the production rate of $Y$ is reduced, leading to a reduction in $Y$ expression levels.

### Simulation of downstream signal response

The model allows calculating the response of the target gene to pre-specified traces of Her6 expression with different dynamic properties (Fig 5A–C), or as a part of an interconnected dynamical system (Fig 5D–F). The interactions governing the dynamics of the target gene are identical in both cases and form a minimal network that is sensitive to the timescale of Her6 fluctuations. We assumed that the downstream target $X$ would be able to self-activate, which is consistent with the literature (Helms *et al*, 2000). We expect other network motifs to have qualitatively similar properties. For example, the self-activation could appear as a consequence of repressing a repressing gene.

We simulate the response of the target gene X to Her6 fluctuations using the differential equation

$$\frac{dX}{dt} = G_1(Y(t)) + G_2(X) - \mu X \tag{2}$$

where $X$ is the protein copy number of the target gene, $t$ is time, $Y(t)$ is copy number of Her6 at time $t$, and $\mu$ is the protein degradation rate of the downstream target gene. The production functions $G_1(Y)$ and $G_2(X)$ describe the repression of $X$ by Her6 and its auto-activatidon, respectively, by defining how the presence of Her6 or

$X$ can influence $X$ production. We chose a production function $G_1(Y)$ that returns a finite production rate, $k_1$, if Her6 copy numbers $Y$ are much lower than a threshold of repression, and which leads to a production rate of 0 if Her6 copy numbers are much greater than this repression threshold. We denote this repression threshold by $Y_0$, which allows us to define

$$G_1(Y) = k_1 \frac{1}{1 + \left(\frac{Y}{Y_0}\right)^n},$$

where $n$ is a Hill coefficient controlling steepness of the transition from $k_1$ to 0 as the Her6 concentration $Y$ increases. The function $G_2(X)$ is defined in similar manner,

$$G_2(X) = k_2 \frac{1}{1 + \left(\frac{X}{X_{0,X}}\right)^{-n}}$$

This function leads to a production rate $k_2$ of $X$ if the number of $X$ molecules is much greater than the activation threshold $X_{0,X}$, and 0 if the number of $X$ molecules is much smaller than the activation threshold. Again, the steepness of the transition is regulated by the Hill coefficient $n$.

In Fig 5A–C, OU samples for $Y$ dynamics were generated first and used as input in equation (2) to simulate the dynamics of $X$. In Fig 5D–F, dynamics of $Y$ and $X$ are simulated jointly by solving the system of stochastic differential equations consisting of equations (1) and (2).

Dynamical systems, such as the ones illustrated in Fig 5A and D and described by equations (1) and (2), may exhibit different qualitative behaviours in different regions of parameter space. To generate Fig 5, we manually selected parameters that place the model in a regime where it exhibits sensitivity to the timescale of the Her6 fluctuations. We used the parameters $k_1 = 0.5$, $k_2 = 10$, $Y_0 = 7.9$, $X_{0,X} = 0.5$, $n = 4$, $\mu = 3$, $\sigma^2 = 1.7$, $Y_{in} = 6.0$, $X_{0,Y} = 2.0$, and three different values for $\alpha_{OU}$ (case 1: $\alpha_{OU} = 2$, case 2: $\alpha_{OU} = 15$, case 3: $\alpha_{OU} = 60$, 100). We implemented the model using a standard forward Euler scheme with a time step of 0.0015 and a simulation duration of 30 time units. Note, however, that the observed qualitative behaviour of the model is not dependent on the specific parameter choices, and similar graphs to those in Fig 5 can be generated with multiple different parameter combinations.

## Data availability

The primary datasets produced in this study are available in the following database:

- Figure source data and datasets: BioStudies, S-BSST374 (https://www.ebi.ac.uk/biostudies/studies/S-BSST374)

**Expanded View** for this article is available online.

## Acknowledgements

We are grateful to Prof. Jon Clarke for help establishing transversal imaging in live Zebrafish and Prof Magnus Rattray for his continued support with statistical data analysis. We kindly thank Dr. Guilherme Costa, Dr. Cerys Manning, Dr.

Tom Pettini, Dr. Thomas Minchington and Dr. Anzy Miller for advice and discussions and Dr. Kyle Wedgewood for bringing inverse stochastic resonance to our attention. The authors would also like to thank the Biological Services Facility, Bioimaging and Systems Microscopy Facilities of the University of Manchester for technical support. This work was supported by a Wellcome Trust Senior Research Fellowship to NP (106185/Z/14/Z). The funders had no role in study design, data collection and analysis, decision to publish or preparation of the manuscript.

## Author contributions

Conceptualization, XS and NP; Methodology, XS and NP; Software, VB and JK; Validation, XS, VB, PD, RL and RT; Formal Analysis, VB, XS and JK; Investigation, XS, RL and PD; Resources, NP, XS and VB; Data Curation, XS and VB; Writing—Original Draft, NP and XS; Writing—Review and Editing, NP, XS, VB and JK; Visualization, XS, VB and JK; Supervision, NP and XS; Project Administration, NP and XS; Funding Acquisition, NP.

## Conflict of interest

The authors declare that they have no conflict of interest.

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
