## [Review Process File · The EMBO Journal]

Dynamic properties of noise and Her6 levels are optimized by miR-9, allowing the decoding of the Her6 oscillator

Ximena Soto, Veronica Biga, Jochen Kursawe, Robert Lea, Parnian Doostdar, Riba Thomas and Nancy Papalopulu

Review timeline:

Submission date:	10th Oct 2019
Editorial Decision:	7th Nov 2019
Revision received:	4th Feb 2020
Editorial Decision:	13th Mar 2020
Revision received:	25th Mar 2020
Accepted:	3rd Apr 2020

Editor: Ieva Gailite

Transaction Report:

1st Editorial Decision

7th Nov 2019

Thank you for submitting your manuscript for consideration by the EMBO Journal. We have now received three referee reports on your manuscript, which are included below for your information.

As you will see from the comments, all reviewers appreciate the work and the topic. However, they also raise a number of substantial and partially overlapping concerns regarding the uncoupling the effect of Her6 expression levels from its expression dynamics, frequency of image collection and technical/biological noise separation that need to be addressed before they can support publication here. From my side, I judge the referee comments to be generally reasonable. Therefore, based on the overall interest expressed in the reports, I would like to invite you to submit a revised version of your manuscript in which you address the comments of all three referees. I should add that it is The EMBO Journal policy to allow only a single major round of revision and that it is therefore important to resolve the main concerns at this stage.

We generally allow three months as standard revision time. Please contact us in advance if you would need an additional extension. As a matter of policy, competing manuscripts published during this period will not negatively impact on our assessment of the conceptual advance presented by your study. However, please contact me as soon as possible upon publication of any related work in order to discuss how to proceed.

REFeree REPORTS:

Referee #1:

The authors address the role of her6 protein dynamics during zebrafish neurogenesis and in particular, investigate the contribution of noise in decoding her6 dynamics. This study builds on a computational approach the authors previously developed (Phillips et al. 2017) that is used to distinguish "periodic (but stochastic) from aperiodic (but fluctuating) phenomena (Phillips et al.

2017)." The authors generate a Her6:Venus knock in fluorescent reporter line using CRISPR/CAS9. They show that during neurogenesis, her6 transitions from aperiodic to oscillatory expression and move on to investigate the role of miR-9 in controlling this transition in her6 dynamics. To this end, they delete a miR-9 binding site within her6 UTR and find that her6 dynamics are classified less frequently as oscillatory and also, that her6 levels are elevated. Furthermore, their analysis reveals an increase in noise as expressed in a lengthening of the aperiodic lengthscale. At the phenotype level, the deletion of miR-9 binding site in her6 leads to an expansion of neural progenitors, while early differentiating neurons were scored absent.

The authors use mathematical modelling to test the effects of altered her6 dynamics and noise on the ability to decode dynamic signals. They propose a model in which depending on noise-levels, a beneficial or detrimental effect on the ability to decode dynamics is observed, the latter termed inverse stochastic resonance effect in analogy to concepts in Engineering and Neuroscience.

This is an elegant study that combines quantitative experimental approaches with sophisticated time-series analysis and mathematical modelling to come up with new conclusions about the role of her6 dynamics and extrinsic noise in the process of neurogenesis. The tools and logic used are analogous to their previous study (Manning et al. 2019,) and the authors find similar trends in gene expression dynamics during neurogenesis. The additional knowledge acquired in a different species corroborates their general model of neurogenesis.

Importantly, this current study performs these quantifications *in vivo*, directly in the context of developing zebrafish embryo, which I think is an important achievement.

One main challenge I see is that the key experimental manipulation presented to test the role of dynamics (i.e. deletion of mir9 binding site) leads to alteration of her6 dynamics but also, at the same time, her6 overall levels change very significantly (see below). It is in my view hence currently not clear how to directly link phenotype to changes in dynamics or even noise. While technically very challenging, experimental strategies to distinguish these effects are needed if conclusions about the role of her6 dynamics are to be made.

Points are detailed below:

1) The deletion of mir9 binding site leads to an "failure to downregulate Her6" (line 334-335) and hence in effect, her6 levels are clearly elevated in MBSm embryos. In parallel, the authors show that her6 dynamics are altered, i.e. oscillations are seen less frequent. It is critical that the effect of elevated her6 levels is addressed and distinguished from effect at the level of her6 dynamics. For instance, a strategy to lower her6 levels in MBSm embryos is needed. Could pharmacological Notch inhibition be applied to MBSm embryos to lower her6 levels? The authors could then directly test if lowering her6 levels leads to a rescue of observed phenotype in MBSm embryos.

2) To support the use of their statistical tools, and the recapitulation of experimental results by their mathematical model, the authors should provide LLR/alpha OU values of simulated trends (results of CTRL parameters vs. MBSm parameters). This would allow quantitative comparison between experimental results and simulations.

3) The authors conclude that the effect of mir9 is at the level of regulating her6 translational efficiency, and not her6 mRNA stability. A direct quantification of her6 mRNA (ISH or better HCR) would provide experimental evidence that can be added to strengthen the argumentation.

- minor concerns that should be addressed

4)(line 311-314) The experiment seems to be done on one embryo per condition (judging from line 812-815.) Although multiple cells are examined within one embryo, due to the variable genotype of MBSm embryos, it would be better to include traces from different embryos.

5)(line 256-261) To the lay reader the connection between the aperiodic lengthscale and noise is not so clear. The logic may be explained in finer detail?

6)(line 302) Sequencing results of CTRL embryos could also be provided. Also, data could be compared to embryos without Cas9 injection for results in Fig3. (Fig S7 is the only direct comparison between uninjected vs. CTRL vs. MBSm embryos, and the differences/similarities are

not so clear.)

FigS1d: GLT is not clear to the reader.

FigS2b: Authors could provide immunostainings of Venus vs. Her6.

Referee #2:

To reveal the dynamics of Her6 expression in neural progenitors, the authors made a Her6-venus knock-in reporter line. They confirmed that the reporter expression represents the endogenous Her6 expression and performed its live imaging in the hindbrain of zebrafish embryos. They found that Her6 expression changes from noisy to oscillatory patterns as neurogenesis progresses, and that Her6 expression oscillates with 1.5-h periodicity. When the miR-9 binding site was deleted from the her6 3'UTR, Her6 protein expression levels increased, making the expression noisier and leading to failure to downregulate Her6. Under this condition, the proneural gene NeuroD4 was downregulated and the transition from progenitor to neuron was impeded, as predicted by mathematical modeling. Based on these results, the authors concluded that noise impairs Her6 oscillation and cell state transition.

This is an interesting work showing the significance of noise, but the authors' imaging method is not appropriate to observe Her6 oscillation. Furthermore, deletion of the miR-9 binding site increased the average level of Her6 expression, and this makes the authors' conclusion ambiguous. Specific comments are indicated below.

Major comments

1. For live imaging of Her6, the authors collected images every 10 min, but this interval may not be appropriate to observe Her6 oscillations. Delaune et al. previously performed live imaging of Her1 by acquiring images every 4 min and detected oscillations with ~30-min periodicity in the presomitic mesoderm of zebrafish embryos. In Fig. 2, the authors concluded that the period of Her6 oscillation is 1.5 h, but this could be much shorter if images were collected every 4 min. It is possible that Her6 expression also oscillates with ~30-min periodicity in neural progenitors, and the authors' method may miss such shorter periodicity. Thus, the authors should collect images with shorter intervals.

2. When miR-9 regulation was removed, Her6 expression became noisier, and the transition from progenitor to neuron was impeded. However, Her6 expression was up-regulated, and it is not clear whether noisier or higher Her6 expression inhibited the transition. To clarify this issue, the authors should test the condition with noisier Her6 expression but without increasing its average level.

Minor comment

3. Fig. 1i indicates that the proportion of Venus+ cells is ~30% at 30hpf, but it looks much less than ~30% in Fig. 1g. The authors should reconfirm the data or provide better images.

Referee #3:

This study of Her6 expression in the developing zebrafish hindbrain aims to link expression dynamics to cell fate transitions. Using a carefully constructed Venus reporter for Her6 protein, the authors show that this construct is both a faithful reporter and does not significantly compromise Her6 function. They then show at a cell population level in rhombomere R6 that there is a pulse of Her6 at 20 hours followed by a linear decay of protein from 25 to 39 hours back to a basal level (figure 1cf). This is corroborated by FCS analysis. The data are thorough and of high quality.

The microRNA miR-9 is a known repressor of Hes like genes and sure enough, Her6 has a single binding site that the authors knocked out using CRISPR. Oddly, the authors rely on analyzing the F0 animals that were injected with Cas9 and gRNA. This will no doubt create variability in F0 genotypes. Each allele will experience a different break-repair and therefore mutation, and some

alleles will not be touched. Off targets will also occur. The authors only genotype 7 F0 animals and find 6 of 7 (85%) have mutations. This is very limited both in terms of numbers (law of large numbers dictates their estimation of 85% is very soft) and information on the nature of the mutations. How many animals have both alleles damaged? One allele? What allelic combinations were created? How mosaic were the animals? - since they used whole animal DNA and chimeric mutagenesis is usual in F0 animals, it is unclear how much of the relevant tissue (R6 hindbrain) was mutant for each. More F0 analysis and better descriptions are required. Alternatively, they should have made a germline mutant and then analyzed a carefully kept and homogeneous genetic stock. Since a large part of the paper relies on mutant analysis, it is disappointing they didn't put the effort into making a proper mutant that they clearly did for the Her6-Venus stock. Genotypic variability translates to phenotypic variability.

In spite of their study of wt Her6 dynamics over the 20 - 40 hour timescale, they fail to do the same for the miR-9 mutants. Instead they really focus on short timescale dynamics (fluctuations). But they should also present a similar cell population scale measurement for Her6 in the mutants from 20 - 40 hours. Fig 3lm hints that the mutants have more Her6 protein in cells at the 48-52 hour time points but it would be helpful to directly compare Her6 dynamics in wt versus mutant to see how the relaxation of Her6 to basal levels is delayed in the mutant on these timescales.

This is important because the cell fate decisions dependent on Her6 are detected only at 52 hours, which is quite late (I assume the 28 hour label in panel of Figure S7d is a typo because in the legend it says 52 hours, like all other sampling). It is very plausible that the delay in longterm relaxation of Her6 by the mutation causes errors in cell fate determination if they are made in that timeframe. Moreover, it speaks to a fundamental weakness of the paper in making the argument that short timescale dynamics occurring between 30-40 hours are responsible for fate decisions or errors.

In other words, how do they know that the mutant's effect on the slow decay of Her6 is not the main driver of cell fate misspecification? Why do they think it is the mutant effect on high frequency oscillations in Her6? Unless they can find a special mutant or condition that uncouples the effects on decay and oscillation, they cannot know which if either is causative. This means they need to rewrite abstract, results and discussion and be completely agnostic about Her6 dynamics and the role it plays in cell fate.

I also have several issues with technical analysis. First, single cell segmentation and tracking. The description of this is ridiculously short given how critical its fidelity must be to a study of noise. Because the noise they measure is a sum of technical noise (segmentation & tracking errors not to mention photon collection and Venus maturation) plus biological noise. They cannot disentangle one from the other easily and if the former outweighs the latter, it becomes difficult to talk with confidence about biological noise. From the Methods, it sounds like it is not real segmentation of bounded objects but Imaris spots. A Gaussian mode? And it sounds like it is not a 3D bounded object but is in 2D. If so, then this is not good. It must be a 3D bounded object to capture all Her6 in a nucleus. Finally, tracking is also an Imaris function and it is completely unclear how well this works. Any segmentation error propagates over time during a movie - so a 1% error rate per frame for a 100 frame movie means that by the end of the movie, ~100% of nuclei are improperly identified. How did they manually correct errors in the segmented time series? All they say is it was manually curated. And finally, did they test their method against some ground truth? Either synthetic data or hand segmented and tracked samples. Much more evidence needs to be provided that this pipeline is acceptable.

They image both Her6 Venus and Histone H2B RFP at the same time in the single cell measures. This is great because RFP could then be used as a constitutive protein control for analysis. However, they do a very puzzling thing. They divide Venus by RFP fluorescence to "normalize" the signal. While such practices are fine for low-tech biology or population level work, it is inappropriate for the type of analysis they perform. Why didn't they also show H2B-RFP in experiments shown in Figure 1d-f? Since they clearly injected the CAAX reporter I assume H2B was also there. What are its long term dynamics?

Coming back to the experiments of Figs 2, 3 - the analysis of single cell dynamics is done with the ratio of Venus/RFP. This needs to instead be done separately on Venus and RFP. Using their opaque Fourier analysis with the ratio (Note - they must provide much much more explanation to the reader

in results and methods precisely what is being done; it's shorthand as is), they see a power spectrum indicating oscillations. They interpret this to be Her6 oscillation. If true, then the Venus analysis alone will also show a similar power spectrum and the RFP will not. In other words, some Her6 fluctuations are oscillatory while H2B fluctuations are stochastic. The power analysis will show that. The authors try and deconstruct using covariance functions but the heart of the matter is Fourier analysis and the results are nicely shown in Figure S5. The authors should move panels e and f from Figure S5 to the main figures! They are the most information rich and decisive, showing aggregate and individual power spectra. The Ornstein Uhlenbeck functions and following statistics are confusing and arbitrary. Classifying a cell into oscillating or not is arbitrary and probably wrong. Virtually every cell had a signal above high frequency power in the ultradian freq range. What is clear is that the mutant cells have an altered behavior. The mutant has diminished power in the lowest frequency cycle. The authors then need to do the same Fourier analysis independently on H2B RFP. Does it only have a spectrum indicating noise? Maybe this is what Tissue Background is in Figure S5e,f. They never state what that is. If it is not H2B, then they should show H2B aggregate and single cell spectra for all data.

I would also think that an autocorrelation analysis might be useful to try. It looks for periodicity without assumptions about its harmonics ($\cos \beta\tau$). Of course the complexity of timescales might mean a signal of periodicity is quite small, but they could compare autocorrelation analysis of Her6 with H2B and see if any region of the curves are different from one another and whether that difference is in the same timescale as the Fourier analysis showed. Comparing mutant and wt this way would also be helpful. An independent method to validate the analysis of oscillations is important.

Were only 14 or 15 cells tracked? What about for experiments in Figure 2? The authors provide no numbers that are easy to find at least.

Finally, I find Figure 5 is one of those "Just so" stories in that lots of modeling will generate something. And sometimes the something can "explain" results. But like a Just so story, this model is ultimately not tested or testable and so it doesn't really mean very much. It should be removed for the sake of clarity and more depth to be added in the remainder of the paper, as outlined.

1st Revision - authors' response

4th Feb 2020

Thank you for giving us the opportunity to reply to the reviewers' comments. We have found the reviewers comments helpful in improving the manuscript. Our detailed point by point response can be found below:

Referee #1:

The authors address the role of her6 protein dynamics during zebrafish neurogenesis and in particular, investigate the contribution of noise in decoding her6 dynamics. This study builds on a computational approach the authors previously developed (Phillips et al. 2017) that is used to distinguish "periodic (but stochastic) from aperiodic (but fluctuating) phenomena (Phillips et al. 2017)." The authors generate a Her6:Venus knock in fluorescent reporter line using CRISPR/CAS9. They show that during neurogenesis, her6 transitions from aperiodic to oscillatory expression and move on to investigate the role of miR-9 in controlling this transition in her6 dynamics. To this end, they delete a miR-9 binding site within her6 UTR and find that her6 dynamics are classified less frequently as oscillatory and also, that her6 levels are elevated. Furthermore, their analysis reveals an increase in noise as expressed in a lengthening of the aperiodic lengthscale. At the phenotype level, the deletion of miR-9 binding site in her6 leads to an expansion of neural progenitors, while early differentiating neurons were scored absent.

The authors use mathematical modelling to test the effects of altered her6 dynamics and noise on the ability to decode dynamic signals. They propose a model in which depending on noise-levels, a beneficial or detrimental effect on the ability to decode dynamics is

observed, the latter termed inverse stochastic resonance effect in analogy to concepts in Engineering and Neuroscience.

We would like to thank the reviewer for reading our paper thoroughly and the nice summary of the work.

This is an elegant study that combines quantitative experimental approaches with sophisticated time-series analysis and mathematical modelling to come up with new conclusions about the role of her6 dynamics and extrinsic noise in the process of neurogenesis. The tools and logic used are analogous to their previous study (Manning et al. 2019,) and the authors find similar trends in gene expression dynamics during neurogenesis. The additional knowledge acquired in a different species corroborates their general model of neurogenesis. Importantly, this current study performs these quantifications in vivo, directly in the context of developing zebrafish embryo, which I think is an important achievement.

One main challenge I see is that the key experimental manipulation presented to test the role of dynamics (i.e. deletion of mir9 binding site) leads to alteration of her6 dynamics but also, at the same time, her6 overall levels change very significantly (see below). It is in my view hence currently not clear how to directly link phenotype to changes in dynamics or even noise. While technically very challenging, experimental strategies to distinguish these effects are needed if conclusions about the role of her6 dynamics are to be made.

This is a valid point that was raised by all reviewers and the Editor- we respond to this criticism at the end of this document so we can address this point collectively.

Points are detailed below:

1) The deletion of mir9 binding site leads to an "failure to downregulate Her6" (line 334-335) and hence in effect, her6 levels are clearly elevated in MBSm embryos. In parallel, the authors show that her6 dynamics are altered, i.e. oscillations are seen less frequent. It is critical that the effect of elevated her6 levels is addressed and distinguished from effect at the level of her6 dynamics. For instance, a strategy to lower her6 levels in MBSm embryos is needed. Could pharmacological Notch inhibition be applied to MBSm embryos to lower her6 levels? The authors could then directly test if lowering her6 levels leads to a rescue of observed phenotype in MBSm embryos.

Please see collective response on levels versus dynamics at the end. Here, we would only mention that our extended time course of Her6 protein quantification by FCS supports our original conclusion that the elevation of Her6 at later stages is due to failure to downregulate it during development and the difference in the level is only 2 fold on average (new Figure 3).

While the Notch inhibitor experiment seems appropriate, there are several issues that make it less useful than one would have thought. First, a Notch inhibitor would affect not only Her6, but also Her9 (mHes1) and Her4 (mHes5) which are also expressed in the hindbrain, and many other genes making any effect non-specific. Second, applying a Notch inhibitor will not help us achieve the fine tuning on Her6 levels needed to achieve a rescue; it is more likely to turn her genes off completely and have a strong phenotype by itself.

2) To support the use of their statistical tools, and the recapitulation of experimental results by their mathematical model, the authors should provide LLR/alpha OU values of simulated trends (results of CTRL parameters vs. MBSm parameters). This would allow quantitative comparison between experimental results and simulations.

We think this refers to the synthetic data in the model in old Figure 5 and note that this section of results has been removed from the manuscript at the request of reviewer 3. Nevertheless the comment is also applicable to the model in new Figure 5 (previously Figure 4a-c) that also includes synthetic Her6 data. We clarify that this mathematical model is not parameterized using experimental data and not intended to be quantitative. The OU

values of simulated traces in Figure 5 are indeed reported accurately, using arrows in Figure 5b and Figure 5e (new result) panels and we have now included the explicit values for α_{OU} also in the Figure legend. This is possible since the aperiodic lengthscale is a parameter of the model. Overall, while there is agreement between model and data we would like to emphasize that the model describes a continuum of possible behaviours and should be viewed qualitatively rather than quantitatively. We have added a statement to this effect in lines 401-404.

3) The authors conclude that the effect of mir9 is at the level of regulating her6 translational efficiency, and not her6 mRNA stability. A direct quantification of her6 mRNA (ISH or better HCR) would provide experimental evidence that can be added to strengthen the argumentation.

This comment refers to the last figure of the paper (computational inference of the likely underlying mechanism) which has been removed at the request of reviewer 3, as it was premature to include it.

- minor concerns that should be addressed

4)(line 311-314) The experiment seems to be done on one embryo per condition (judging from line 812-815.) Although multiple cells are examined within one embryo, due to the variable genotype of MBSm embryos, it would be better to include traces from different embryos.

The single cell live imaging we are employing imposes restrictions on the number of embryos we can analyse per experiment. Nevertheless, we have now included additional traces from more MBSm and CTRL embryos in separate experiments (new Figure EV3). We have also included traces that have been acquired on a different microscope to further exclude technical artefacts (Materials and methods-Live imaging for single cell tracking). In response to reviewer 3, we have also increased the sample size of the sequencing analysis to show the high prevalence of mutagenesis (Figure S4b,c).

5)(line 256-261) To the lay reader the connection between the aperiodic lengthscale and noise is not so clear. The logic may be explained in finer detail?

We have rephrased this as “A higher lengthscale indicates that subsequent points in a time trace become un-correlated faster (e.g. a decay in signal autocorrelation) and is therefore used here as a measure of noise in a dynamic trace” (lines 247-250). To make it more intuitive to understand we have expanded the number of examples of traces with different α_{OU} value (previously Figure 4f-l and new examples in Figure EV2). Since the estimate of aperiodic lengthscale is subject to assumptions made in the OU covariance model, we validated that increased values are indicative of high frequency noise by independent measurements from the power spectrum analysis, i.e. a non-parametric technique. This is found in the revised manuscript in lines 340-349, 356-361 and **Figure EV4d-g**.

6)(line 302) Sequencing results of CTRL embryos could also be provided. Also, data could be compared to embryos without Cas9 injection for results in Fig3. (Fig S7 is the only direct comparison between uninjected vs. CTRL vs. MBSm embryos, and the differences/similarities are not so clear.)

Sequencing of CTRL embryos has now been included in **Figure S4b**. We have compared the dynamic parameters between NPCs from embryos w/o Cas9 sgRNA injection and control (CTRL) embryos at 34hpf and found no change in number of oscillators, aperiodic lengthscale and period. Essentially, we find that CTRL behaves the same as WT. This has been added to the manuscript in new **Figure S4d,e,f**. We have also added a sentence to reflect this in the manuscript: “As expected, NPCs from CTRL embryos recapitulated Her6

dynamics observed in embryos without Cas9/sgRNA (Figure S4d-f) and presence of oscillatory Her6 activity (Figure EV3a)" (lines 334-336).

FigS1d: GLT is not clear to the reader. We have defined GLT as germ line transmission in the legend of **Figure S1**.

FigS2b: Authors could provide immunostainings of Venus vs. Her6. We regret that we have not been able to do this because there is no antibody sensitive enough available for her6 immunostaining.

Referee #2:

To reveal the dynamics of Her6 expression in neural progenitors, the authors made a Her6-venus knock-in reporter line. They confirmed that the reporter expression represents the endogenous Her6 expression and performed its live imaging in the hindbrain of zebrafish embryos. They found that Her6 expression changes from noisy to oscillatory patterns as neurogenesis progresses, and that Her6 expression oscillates with 1.5-h periodicity. When the miR-9 binding site was deleted from the her6 3'UTR, Her6 protein expression levels increased, making the expression noisier and leading to failure to downregulate Her6. Under this condition, the proneural gene NeuroD4 was downregulated and the transition from progenitor to neuron was impeded, as predicted by mathematical modeling. Based on these results, the authors concluded that noise impairs Her6 oscillation and cell state transition.

This is an interesting work showing the significance of noise, but the authors' imaging method is not appropriate to observe Her6 oscillation. Furthermore, deletion of the miR-9 binding site increased the average level of Her6 expression, and this makes the authors' conclusion ambiguous. Specific comments are indicated below.

Thank you for the nice summary of the work; we reply to the concerns below.

Major comments

1. For live imaging of Her6, the authors collected images every 10 min, but this interval may not be appropriate to observe Her6 oscillations. Delaune et al. previously performed live imaging of Her1 by acquiring images every 4 min and detected oscillations with ~30-min periodicity in the presomitic mesoderm of zebrafish embryos. In Fig. 2, the authors concluded that the period of Her6 oscillation is 1.5 h, but this could be much shorter if images were collected every 4 min. It is possible that Her6 expression also oscillates with ~30-min periodicity in neural progenitors, and the authors' method may miss such shorter periodicity. Thus, the authors should collect images with shorter intervals.

We are sorry for the misunderstanding of how images were collected. Single cell traces to analyse oscillation we collected at 6min intervals (not 10 minute intervals as the reviewer suggests) therefore it is quite comparable to the Delaune et al study (Delaune, Francois et al., 2012). The her6 reporter is a knock-in of the endogenous copy and hence most likely less bright than the Her1 transgene that was used in the Delaune study (which was most likely overexpressed as evidenced by the fact that there was phenotype in the homozygous fish). We find that bleaching of the low abundance Her6::Venus reporter prevented us from using even shorter imaging intervals. We have stated the acquisition interval in the main text (lines 216-217) and in the amended methods (Materials and Methods- Live imaging for single cell tracking). We should also point out that there is heterogeneity in the periodicity detected and a few cells in the early stages of 28 and 30hpf have periodicity around 30 mins (see **Figure 2g, 28hpf** and **Figure EV1a**). This shows that we are able to detect periodicities of 30 mins but these are rare. Revealing the single cell heterogeneity that exists in the tissue is part of the strength of our analysis.

2. When miR-9 regulation was removed, Her6 expression became noisier, and the transition from progenitor to neuron was impeded. However, Her6 expression was up-regulated, and it is not clear whether noisier or higher Her6 expression inhibited the transition. To clarify this issue, the authors should test the condition with noisier Her6 expression but without increasing its average level.

This valid point was raised by 2 reviewers and the editor, therefore we will address all comments together at the end of this rebuttal document. We would nevertheless like to emphasise here that the apparent Her6 upregulation is in fact a failure to downregulate the early levels. Her6 is never upregulated above its normal earlier development levels. This is shown by the extended protein abundance data by FCS presented in the new **Figure 3f,g**.

Minor comment

3. Fig. 1i indicates that the proportion of Venus+ cells is ~30% at 30hpf, but it looks much less than ~30% in Fig. 1g. The authors should reconfirm the data or provide better images.

There is heterogeneity at the level of her6 expression seen in snapshots images. The calculation of 30% Venus+ cells includes high as well as low intensities, We provide a higher brightness image to show this more convincingly and underlined the area that expresses Her6::Venus. This is shown in new **Figure 1g**.

Referee #3:

This study of Her6 expression in the developing zebrafish hindbrain aims to link expression dynamics to cell fate transitions. Using a carefully constructed Venus reporter for Her6 protein, the authors show that this construct is both a faithful reporter and does not significantly compromise Her6 function. They then show at a cell population level in rhombomere R6 that there is a pulse of Her6 at 20 hours followed by a linear decay of protein from 25 to 39 hours back to a basal level (figure 1cf). This is corroborated by FCS analysis. The data are thorough and of high quality.

Thank you for finding the data thorough and of high quality.

The microRNA miR-9 is a known repressor of Hes like genes and sure enough, Her6 has a single binding site that the authors knocked out using CRISPR. Oddly, the authors rely on analyzing the F0 animals that were injected with Cas9 and gRNA. This will no doubt create variability in F0 genotypes. Each allele will experience a different break-repair and therefore mutation, and some alleles will not be touched. Off targets will also occur. The authors only genotype 7 F0 animals and find 6 of 7 (85%) have mutations. This is very limited both in terms of numbers (law of large numbers dictates their estimation of 85% is very soft) and information on the nature of the mutations. How many animals have both alleles damaged? One allele? What allelic combinations were created? How mosaic were the animals?

- since they used whole animal DNA and chimeric mutagenesis is usual in F0 animals, it is unclear how much of the relevant tissue (R6 hindbrain) was mutant for each. More F0 analysis and better descriptions are required. Alternatively, they should have made a germline mutant and then analyzed a carefully kept and homogeneous genetic stock. Since a large part of the paper relies on mutant analysis, it is disappointing they didnt put the effort into making a proper mutant that they clearly did for the Her6-Venus stock. Genotypic variability translates to phenotypic variability.

We agree with the reviewer's comment that a germline mutant would be less heterogeneous. We regret that we do not have a germline mutant that we can use within the time framework of this study, bearing in mind that generating the Knock-in fish took 2 years and the overall work described in this paper needed 4 years to complete. Having said that, CRISPR mutagenesis (InDel) is very efficient (unlikely the generation of in frame

Knock-ins) and it is therefore possible to obtain meaningful results based on transient injection. The purpose of our mutagenesis is to interfere with the miR-9 binding site therefore the accuracy of mutagenesis is less important in this particular context, as long as the miR-9 binding site is mutated. The injected Zebrafish are indeed mosaic, therefore, we have used H2B-mKeima/CAAX-mRFP co-injection with gRNA/Cas9 in order to identify embryos that show expression in the area of interest (R5/6) and with low mosaicism. Thank you for giving us the alternative option of increasing the sample size. Indeed, to increase confidence in the miR-9 mutagenesis we have increased the number of samples that we have analysed for mutations. Briefly, we have analysed by PCR 8 clones from each of 10 injected embryos. Following sequencing, 80% of those carried an insert with mutation and of those 100% were mutated at the miR-9 binding site (mainly deletions). This has been made clear in the main text (lines 311-313). The additional sequencing data has been added in **Figure S4b,c** and the procedure is now better explained in the Materials and methods-Microinjection and genotyping.

In spite of their study of wt Her6 dynamics over the 20 - 40 hour timescale, they fail to do the same for the miR-9 mutants. Instead they really focus on short timescale dynamics (fluctuations). But they should also present a similar cell population scale measurement for Her6 in the mutants from 20 - 40 hours.

We have now done this and it is shown in new **Figure 3**. The new imaging and protein abundance analysis gives an overview of Her6 expression in the Hindbrain in Control (CTRL) versus mutant (MBSm) embryos over development.

Fig 3Im hints that the mutants have more Her6 protein in cells at the 48-52 hour time points but it would be helpful to directly compare Her6 dynamics in wt versus mutant to see how the relaxation of Her6 to basal levels is delayed in the mutant on these timescales.

We have extended the absolute quantification of the Her6 protein by FCS in single cells from CTRL and mutant embryos from 34hpf to 69hpf. Specifically, we repeated the FCS experiment so we have added more samples in the early time points and extended the time course to later time points. The results are shown in new **Figure 3f-g**. The results show some cells drop the Her6 levels towards basal levels at later time points, suggesting that indeed the relaxation of Her6 levels is delayed. Importantly, this result also shows that the median Her6 expression level is never above that of control at 34hpf. Our interpretation of this result is that Her6 *fails to decline* at the same rate as the control, rather than being genuinely upregulated. This is discussed in the manuscript at lines 317-326.

This is important because the cell fate decisions dependent on Her6 are detected only at 52 hours, which is quite late (I assume the 28 hour label in panel of Figure S7d is a typo because in the legend it says 52 hours, like all other sampling). It is very plausible that the delay in longterm relaxation of Her6 by the mutation causes errors in cell fate determination if they are made in that timeframe. Moreover, it speaks to a fundamental weakness of the paper in making the argument that short timescale dynamics occurring between 30-40 hours are responsible for fate decisions or errors.

Figure S7d, is correctly labelled as 28hpf however we apologise for the confusion. Just to clarify **Figure S7d** shows that there is no phenotype at 28hpf; this is because miR-9 is not expressed at this stage in the hindbrain (see new **Figure 3a**). Therefore, we would not expect a phenotype at 28hpf and the figure is essentially a control. It also shows that there are no off-target effects in the hindbrain as development is normal at this stage, as judged by *gfap*, *elavl3* and *her6* expression. The phenotype becomes apparent later, when miR-9 would normally start to be expressed. Thus, the time of analysis of the phenotype, the appearance of the phenotype and the expression of miR-9 are consistent with each other. We do not see where there is a fundamental weakness here. The figure content of S7 has been re-arranged and this is now found in **Figure S5c-e**.

In other words, how do they know that the mutant's effect on the slow decay of Her6 is not

the main driver of cell fate misspecification? Why do they think it is the mutant effect on high frequency oscillations in Her6? Unless they can find a special mutant or condition that uncouples the effects on decay and oscillation, they cannot know which if either is causative. This means they need to rewrite abstract, results and discussion and be completely agnostic about Her6 dynamics and the role it plays in cell fate.

Uncoupling the effects of level from dynamics is a valid point, raised by all reviewers and the Editor. We have made some key discoveries on this front and we address it separately at the end of this document.

I also have several issues with technical analysis. First, single cell segmentation and tracking. The description of this is ridiculously short given how critical its fidelity must be to a study of noise.

We apologise that the details were not sufficient; we have now made the description more detailed and easier to follow (Materials and Methods-Single Cell Tracking).

Because the noise they measure is a sum of technical noise (segmentation & tracking errors not to mention photon collection and Venus maturation) plus biological noise. They cannot disentangle one from the other easily and if the former outweighs the latter, it becomes difficult to talk with confidence about biological noise.

The reviewer is highlighting an issue which is indeed a challenge. We have taken several steps aimed to address it at all levels in the analysis:

- Imaging: acquisition parameters are optimised independently for each channel to minimise bleaching and avoid saturation while retaining single cell resolution (outlined in Materials and Methods- Live imaging for single cell tracking);
- Segmentation and tracking: we give particular attention to correct positioning of the 3D spot at the center of the nucleus in each frame. We have tested the error between the manually tracked 3D spot and the center of mass obtained from automated segmentation and found it to be less than 0.5um on average per track (see **Figure S3b**). We also manually curate all tracks to avoid propagation of errors and this is now explained more in detail in Materials and Methods-Single Cell Tracking.
- Technical noise: we obtain an experimental readout of noise from the detector by using a 3D spot tracking in an area of the tissue that does not express the fluorophore and we refer to this background (Materials and Methods- Background fluorescence and example in new Figure S4d). We do this independently for each channel and as expected, we find that the background signal behaves like low power white noise (see new Figure EV2 and EV4).
- Dynamic parameter estimation from data: we use statistics from background traces in the estimation of dynamic parameters from single cell data (now found in Figure 2 and Figure 4) and we do so independently per experiment. Specifically, we impose that the residual signal representing the difference between detrended data and the covariance models is randomly distributed with the same variance as the technical noise from background traces. This prevents overfitting and interpreting technical fluctuations as noise. We have explained this in better detail in Material and Methods-Dynamic data analysis- Calibration of technical noise from background.
- Biological noise: as the reviewer pointed out, we have mKeima-H2B expressed in the same cells which enables us to have a measure of biological noise at single cell level. We apologise for not using this more effectively in the initial submission. To alleviate reviewers' concerns, we have included the analysis of H2B fluctuations with the same pipeline and report minimal false positive rates (4.5% of H2B traces are marked as oscillatory, see new Figure S4e). We have also analysed the power spectrum of H2B traces compared to Venus traces and found that H2B

has a lower power and lower coherence ultradian peak compared to Venus and Venus/H2B ratio (Figure EV2a,b). This highlights the important observation that H2B is not white noise and we cannot interpret the presence of an ultradian peak in the power spectra as proof of oscillatory activity (see Figure EV2b and Figure EV4e). In addition, we incorporated more examples of H2B in the same cells in new Figure EV3. Finally, we report no change in aperiodic lengthscale of mKeima-H2B time series between CTRL and MBSm (new Figure S4h,i).

We have clarified this in the appropriate results and methods sections.

From the Methods, it sounds like it is not real segmentation of bounded objects but Imaris spots. A Gaussian mode? And it sounds like it is not a 3D bounded object but is in 2D. If so, then this is not good. It must be a 3D bounded object to capture all Her6 in a nucleus.

The imaging analysis is 3D, using the Imaris "Spot" function in 3D mode. Thus, the intensity is tracked with a 3D object (a ball) placed manually in the middle of the nucleus. The images are captured with sufficient z-depth to capture full cells in 3D. The size of the ball covers approximately 75% of the nuclear volume. Due to technical limitations it is not possible currently to accurately track the entire nucleus. We have clarified this in the technical description of the tracking in Materials and Methods- Single Cell Tracking.

Finally, tracking is also an Imaris function and it is completely unclear how well this works. Any segmentation error propagates over time during a movie - so a 1% error rate per frame for a 100 frame movie means that by the end of the movie, ~100% of nuclei are improperly identified. How did they manually correct errors in the segmented time series? All they say is it was manually curated. And finally, did they test their method against some ground truth? Either synthetic data or hand segmented and tracked samples. Much more evidence needs to be provided that this pipeline is acceptable.

We agree that using Imaris with no manual correction might have given rise to a lot of error in the analysis. However, we do go back and manually check every single frame, of every single track. This means that everything is effectively hand segmented, which is laborious and slow but necessary to ensure that there are as few errors as possible. However, we find that when cells are isolated and thus quite easily distinguishable from neighbouring nuclei there is good correspondence between automatic and manual tracking and examples have been added to **Figure S3b**. We have made this clearer in the Materials and Methods-Single Cell Tracking.

They image both Her6 Venus and Histone H2B RFP at the same time in the single cell measures. This is great because RFP could then be used as a constitutive protein control for analysis. However, they do a very puzzling thing. They divide Venus by RFP fluorescence to "normalize" the signal. While such practices are fine for low-tech biology or population level work, it is inappropriate for the type of analysis they perform. Why didn't they also show H2B-RFP in experiments shown in Figure 1d-f? Since they clearly injected the CAAX reporter I assume H2B was also there. What are its long term dynamics?

Apologies of the omission of more mKeima-H2B traces which have now been included in new Figure EV3 in addition to previous ones (Figure 2d and Figure EV1). H2B-mKeima traces sometimes show spikes, but they are random and the tracks do not pass an oscillatory test (new Figure S4d). This is reassuring as the behaviour is clearly distinct from Her6-Venus (new Figures EV1 to EV4). We would expect that the mKeima-H2B signal is eventually bleached and also we would expect injected mRNA degradation at late development stages (after 48hpf), as such its long term dynamics are not biologically relevant.

We divide the Her6::Venus traces by mKeima-H2B in order to remove any fluctuations which are due to non-specific noise (be it technical or simply unrelated to the mechanism that generates her6 oscillations), Similar normalisation has been performed by other single

cell state-of-the art dynamic papers, such as the recent (Yoshioka-Kobayashi, Matsumiya et al., 2020).

Coming back to the experiments of Figs 2, 3 - the analysis of single cell dynamics is done with the ratio of Venus/RFP. This needs to instead be done separately on Venus and RFP. Using their opaque Fourier analysis with the ratio (Note - they must provide much more explanation to the reader in results and methods precisely what is being done; it's shorthand as is), they see a power spectrum indicating oscillations. They interpret this to be Her6 oscillation. If true, then the Venus analysis alone will also show a similar power spectrum and the RFP will not. In other words, some Her6 fluctuations are oscillatory while H2B fluctuations are stochastic. The power analysis will show that. The authors try and deconstruct using covariance functions but the heart of the matter is Fourier analysis and the results are nicely shown in Figure S5. The authors should move panels e and f from Figure S5 to the main figures!

We have taken onboard the reviewer's suggestion and we are presenting power spectrum analysis of individual and normalized signals. Briefly the analysis supported our conclusions; we thank the reviewer for these comments that resulted in more robust analysis of our data. Specifically, we have found no differences when analysing Her6::Venus or Her6::Venus/H2B in knock-in embryos at different stages in development using power spectrum analysis (included in new **Figure EV2**). Both of these signals show very similar spectral characteristics showing that we do not "lose" oscillators by normalising in this way. Furthermore, the results support our finding that the number of oscillators increases at 30hpf and 34hpf compared to 28hpf. Furthermore, the peak of the power spectra of Venus coincides with Venus/H2B and agree very well with our estimate of 1.5h. Specifically, we confirm that when oscillators are prevalent Her6/H2B shows high coherence levels similar to Venus and consistently above H2B (**Figure EV2**, 30hpf and 34hpf) whereas when the number of oscillators is low coherence is comparable to H2B coherence (**Figure EV2**, 28hpf). We have also included a power spectrum analysis of H2B for the CTRL versus MBSm data (now included in **Figure EV4**). Here we also find that coherence of data from CTRL embryos where oscillators are frequently detected is higher compared to coherence of H2B whereas this reduces in MBSm embryos where we reported a reduction in oscillatory activity (new **Figure EV4c**). These results provide an alternative method to validate the presence of oscillations at population level and have been included in the manuscript at lines 259-267 and lines 340-349.

They are the most information rich and decisive, showing aggregate and individual power spectra. The Ornstein Uhlenbeck functions and following statistics are confusing and arbitrary. Classifying a cell into oscillating or not is arbitrary and probably wrong. Virtually every cell had a signal above high frequency power in the ultradian freq range. What is clear is that the mutant cells have an altered behavior. The mutant has diminished power in the lowest frequency cycle. The authors then need to do the same Fourier analysis independently on H2B RFP. Does it only have a spectrum indicating noise? Maybe this is what Tissue Background is in Figure S5e,f. They never state what that is. If it is not H2B, then they should show H2B aggregate and single cell spectra for all data.

I would also think that an autocorrelation analysis might be useful to try. It looks for periodicity without assumptions about its harmonics ($\cos \beta\tau$). Of course the complexity of timescales might mean a signal of periodicity is quite small, but they could compare autocorrelation analysis of Her6 with H2B and see if any region of the curves are different from one another and whether that difference is in the same timescale as the Fourier analysis showed. Comparing mutant and wt this way would also be helpful. An independent method to validate the analysis of oscillations is important.

To increase confidence in our method we have verified the data with a different method. The following points can be made in detail; first, while 60 to 80% of tracks can be classified as oscillatory based on our covariance function method, only 4.5% of the mKeima-H2B signal passes this test.

The result is similar when the data is analysed with a different methods (Lomb-Scargle Periodogram, LSP) although by comparison it is clear that LSP classifies over 30% of H2B tracks are oscillatory whereas Venus signals show higher rates of passing cells (see graph below). We note however that the LSP method allows too many false positives through in order to be used reliably, which is consistent with what the findings reported in our earlier paper (Phillips, Manning et al., 2017). As we state in our previous paper our method uses synthetic OU data to be well calibrated. The second point is that traces obtained by H2B also show a peak in the power spectrum, albeit a lower coherence peak, indicating that H2B is not white noise (and perhaps we would not expect it to be as transcription/translation can be bursty with some frequency). Thus, it is very difficult to accurately classify a noisy oscillator based on the presence of a peak in the power spectrum. This is exactly the reason that we have developed a covariance model method for such data.

In addition, we have done the Power spectrum analysis that the reviewer suggested and the results support our initial conclusions (see new **Figure EV2** and **EV4**). There is no perfect method for the analysis of such noisy and sparse data; each has their own advantages and limitations, and the fact that a number of different analysis pipelines support our conclusion increases confidence to our interpretation of the data.

We are sorry to hear that the reviewer found the Ornstein-Uhlenbeck method confusing. We have tried to clarify this methodological section by rewriting it (see Materials and Methods- Dynamic data analysis). Having said that, we would like to defend our method, which has been peer-reviewed, published in PLOS computational Biology as a method (Phillips et al., 2017) and subsequently used to analyse data in (Manning, Biga et al., 2019), with some modifications. We believe we have tested it extensively, both in terms of controls and in testing it alongside other methods. Therefore, we have kept it in the main figures while the power analysis has now been elevated to expanded view Figures EV2 and EV4.

Were only 14 or 15 cells tracked? What about for experiments in Figure 2? The authors provide no numbers that are easy to find at least.

We have included detailed sample sizes with numbers of cell tracked and numbers of embryos in the appropriate figure legends.

Finally, I find Figure 5 is one of those "Just so" stories in that lots of modeling will generate something. And sometimes the something can "explain" results. But like a Just so story, this model is ultimately not tested or testable and so it doesn't really mean very much. It

should be removed for the sake of clarity and more depth to be added in the remainder of the paper, as outlined.

We think that implications of the computational modelling in old Figure 5 are very interesting but we do agree that including it in this manuscript is premature, as we don't have yet experimental data to fully test the predictions. Thus, we have taken the reviewers suggestion and we have removed it, reserving it for future work. This also allowed us more space to deepen on the modelling in new **Figure 5** (previously Figure 4). We think this strengthened the manuscript and we thank you for the suggestion.

Response to collective criticism on “Her 6 levels” versus “Her 6 dynamics” issue

All reviewers expressed some scepticism as to whether the observed phenotype is due to “elevated” her6 level versus the change in her6 dynamics. We recognise this as a valid criticism and we have taken several steps to address it.

To summarise the issue here, we report that mutagenising the miR-9 binding site in the 3' UTR of her6, results in failure to downregulate Her6 protein levels during development, in less Her6 time traces passing an oscillatory test, in an increase in noise with high frequency characteristics and in a failure to upregulate downstream targets of Her6 that mark/control differentiation such as elav3 and neuroD4.

To distinguish the effects of a change in protein level versus a change in protein dynamics, ideally, one would have liked an experiment where these can be changed independently, with a high level of control, to mimic the experimentally observed differences in levels and dynamics. However, we are not aware of an experimental design that will allow such exquisite control and it is something that has never been done before. We do not think it is possible with current technologies and we note that the reviewers do not have specific suggestions.

We argue here that separating these effects experimentally may not even be possible, if the change of level and the change in dynamics are mechanistically linked. For example, if a change in Her6 dynamics *causes* a change in Her6 levels, then changing the dynamics will also change the levels. In addition, in this scenario where the change in the level is downstream to the change in dynamics, a change in levels will have the same effect as the change in dynamics. It follows that lowering the level of Her6 with a DAPT inhibitor (see additional problems above) or by overexpressing it, will almost certainly have an effect but it will not tell us what the sequence of molecular events in normal development is.

In the revised manuscript, we provide additional evidence to support our suggestion that the change in Her6 dynamics precedes and underlies a change in the level of Her6 and is by itself sufficient to cause a phenotype. We have previously shown by computational modelling that changing the frequency of her6 dynamics (input Y), impedes the upregulation of downstream her6 targets (target X) (new **Figure 5a-c**). In that model, Y (Her6) was not downregulated, supporting the idea that changing Her6 dynamics alone is sufficient to prevent upregulation of downstream targets. To make the model better aligned with the in vivo observations, we have now added negative feedback from the downstream target towards Her6. Such dual negative feedback is very common in development. This revised model shows that, in this case, her6 dynamics also fail to be downregulated when the her6 input changes faster, but that this is a consequence of failing to upregulate the downstream target (new Figure 5d-f and lines 390-396). Importantly, her6 levels are not elevated in this model, they just fail to be downregulated, which fits very well with the experimental FCS data (new **Figure 3**).

Thus, the order of events in this cell state transition appear to be that when Her6 dynamics change to high frequency input, downstream targets fail to be upregulated and as a consequence Her6 fails to be downregulated.

To provide experimental support for this model we have examined the expression of her6 and elav3, with double smFISH probes, at single cell resolution. If our model is correct, we

would find that her6 and elav3 are co-expressed, because elav3 would be activated before her6 is downregulated. This is exactly what we see in the data presented in **Figure 6b** and discussed in lines 426-432.

We believe this is highly significant not only because it supports our model but because it provides for the first-time insight into the events that lead into a cell state transition. We argue that when the transition has been made, the level of her6 is downregulated but this is a consequence of the cell state transition having been made rather than the cause.

We thank the reviewers for motivating us to do the extra work that was required to strengthen our model. At the same time, we have made it clear in the text that future experimental evidence will be needed to fully test this model and alternative interpretations are still possible. In line with softening this conclusion we have also changed the title of the manuscript from “Dynamic properties of noise prevent the decoding of Her6 oscillator through inverse stochastic resonance” to “Dynamic properties of noise and her6 levels are optimized by miR-9, allowing the decoding of the Her6 oscillator”.

References

- Delaune EA, Francois P, Shih NP, Amacher SL (2012) Single-cell-resolution imaging of the impact of Notch signaling and mitosis on segmentation clock dynamics. *Dev Cell* 23: 995-1005
- Manning CS, Biga V, Boyd J, Kursawe J, Ymisson B, Spiller DG, Sanderson CM, Galla T, Rattray M, Papalopulu N (2019) Quantitative single-cell live imaging links HES5 dynamics with cell-state and fate in murine neurogenesis. *Nature Communications* 10: 2835
- Phillips NE, Manning C, Papalopulu N, Rattray M (2017) Identifying stochastic oscillations in single-cell live imaging time series using Gaussian processes. *PLoS Comput Biol* 13: e1005479
- Yoshioka-Kobayashi K, Matsumiya M, Niino Y, Isomura A, Kori H, Miyawaki A, Kageyama R (2020) Coupling delay controls synchronized oscillation in the segmentation clock. *Nature*

2nd Editorial Decision

13th Mar 2020

Thank you for submitting a revised version of your manuscript. I sincerely apologise for the delay in communicating the decision. I have now received reports from two of the original referees, who find that their main concerns have been addressed and are now in broadly favour of publication of the manuscript. There now remain only a few mainly editorial issues that have to be addressed before I can extend formal acceptance of the manuscript.

REFEREE REPORTS:

Referee #2:

The authors have properly addressed most of my concerns, and now I have further comments.

Referee #3:

The authors did a fine job in dealing with almost all of the issues I raised. The one exception is the issue that the authors rebut at the end of their responses because all 3 reviewers had the same issue.

They have not adequately provided evidence to show the miRNA's effects on high frequency noise or oscillations are responsible for the developmental errors they observe in the miR-9 mutant fish.

They argue that new double-triple label smFISH experiments support their favoring the high frequency model. The experiments show that elav and her6 RNAs are present together in a subset of cells. They argue that the presence of such cells supports the high frequency model. I do not agree.

First, it is conflating nascent RNAs with protein measurements of Her6 done in the other experiments. RNAs typically have short half-lives while proteins do not. Second, there is always a timelag between RNA transcription and protein activity. Fixed imaging of nascent RNAs cannot detect Her6 protein and elav RNA together.

Third, even if there were Her6 protein and elav RNAs co-existing in the same cells, it does not rule out longer timescale interactions. Elav is not a "marker" of a cell state transition. They use the words marker and transition loosely, and given the rigor of the manuscript, they should not do so. Although elav expression stays at a high steady state in neurons, the initial transient establishment of that expression is not necessarily a precise dividing line between one cell state and another. Cells are not discretized in time - state transitions are continuous. They can be rapid but they are not instantaneous. There are many examples in the Drosophila literature (embryos and imaginal discs) where opposing fate determinants are transiently co-expressed in cells, which then resolve over time into a final state having one or the other determinant.

Such feedback systems can act over long timescales, and microRNAs increase the time it takes for the level of the upstream protein to relax to a basal low steady state. Please see Cassidy et al, (2019 Cell 178, 980) for the use of control theory to model such regulatory feedback and show that microRNAs do indeed accelerate the relaxation - precisely what the authors see with Her6 and miR-9 in Fig 5f,g

The authors argue in their general response to all reviewers that it is extremely difficult to parse between high frequency dynamics and longer timescale dynamics for finding the cause of the mutant phenotypes. I completely agree. I think it is unnecessary to have such evidence and publish this work. I think it is necessary to be utterly agnostic about what might be the cause. Instead, devote the discussion to exploring each feature (noise, oscillations, relaxation) and how each might influence fate transitions. They should delete the sections in the abstract results and discussion that unjustifiably push their favored mechanism. The evidence is not strong enough to validate it. It will be a better and more thought provoking paper if it stimulates the field to consider a complex problem rather than a simple but potentially incorrect one.

2nd Revision - authors' response

25th Mar 2020

The authors performed the requested editorial changes.

3rd Editorial Decision

3rd Apr 2020

Thank you for addressing the final minor issues. I am now pleased to inform you that your manuscript has been accepted for publication in The EMBO Journal.

Corresponding Author Name: Soto X and Papalopulu N

Manuscript Number: EMBOJ-2019-103558R